# DE-MARK: Watermark Removal in Large Language Models

Ruibo Chen [*1]   Yihan Wu [*1]   Junfeng Guo [1]   Heng Huang [1]

## Abstract

Watermarking techniques offer a promising way to identify machine-generated content via embedding covert information into the contents generated from language models (LMs). However, the robustness of the watermarking schemes has not been well explored. In this paper, we present DE-MARK, an advanced framework designed to remove n-gram-based watermarks effectively. Our method utilizes a novel querying strategy, termed random selection probing, which aids in assessing the strength of the watermark and identifying the red-green list within the n-gram watermark. Experiments on popular LMs, such as Llama3 and ChatGPT, demonstrate the efficiency and effectiveness of DE-MARK in watermark removal and exploitation tasks.

## 1. Introduction

As language models rapidly evolve and their usage becomes more widespread (Wang et al., 2024a; Thirunavukarasu et al., 2023; Rubenstein et al., 2023; Wang et al., 2024b; Kasneci et al., 2023; Wang et al., 2023a; Chen et al., 2024b), concerns among researchers and regulators about their potential for misuse, particularly in generating harmful content, have intensified. The challenge of distinguishing between human-written and AI-generated text has emerged as a key focus of research. Statistical watermarking (Aaronson, 2022; Kirchenbauer et al., 2023a) presents a promising approach to identifying text produced by sequential LLMs. This method embeds a statistical pattern into the generated text via a pseudo-random generator, which can later be detected through a statistical hypothesis test.

However, recent research (Jovanović et al., 2024) highlights that the robustness of existing n-gram watermarking schemes remains insufficiently understood and, in many cases, significantly overestimated. One critical vulnerability lies in watermark stealing & removal, a method that seeks to reverse-engineer the watermarking process. With this method, adversaries can effectively strip or alter the embedded watermark, undermining its intended protection. This poses a serious challenge to the security and integrity of watermarked content, as it exposes a weakness in the watermarking scheme's ability to withstand tampering.

In this paper, we delve deeper into watermark removal techniques and introduce DE-MARK, a comprehensive framework for stealing and removing n-gram-based watermarks. Our approach utilizes a querying strategy on the language model to reconstruct the red and green lists and estimate the watermark strength. Unlike prior work (Jovanović et al., 2024), which has certain limitations – namely, a) requiring knowledge of the underlying hash function of the n-gram and b) requiring paraphrase tool for watermark removal, which cannot preserve the original LM distribution – our method is more general and provides a theoretical guarantee between the post-removal LM distribution and the original ones. Additionally, DE-MARK can be adapted for watermark exploitation by imitating the watermarking rule and employing another language model to generate watermarked content.

Our contribution can be summarized as follows:

- We propose DE-MARK, a comprehensive watermark framework designed for the removal of n-gram watermarks (Kirchenbauer et al., 2023a), which is considered a pivotal approach in the field. Our framework introduces a provable unbiased estimator to assess the strength of watermarks and offers theoretical guarantees about the gap between the original and the post-removal distributions of the language model. Essentially, DE-MARK operates without requiring prior knowledge of the n-gram watermark parameters.

- Through extensive experiments, we demonstrate the efficacy of DE-MARK in watermark removal and exploitation tasks from well-known language models, such as Llama3 and Mistral. Additionally, a case study on ChatGPT further confirms DE-MARK's capability to effectively remove watermarks from industry-scale

---

[*]Equal contribution [1]Department of Computer Science, University of Maryland, College Park. Correspondence to: Ruibo Chen <rbchen@umd.edu>, Yihan Wu <ywu42@umd.edu>, Junfeng Guo <gjf2023@umd.edu>, Heng Huang <heng@umd.edu>.

*Proceedings of the 42nd International Conference on Machine Learning*, Vancouver, Canada. PMLR 267, 2025. Copyright 2025 by the author(s).

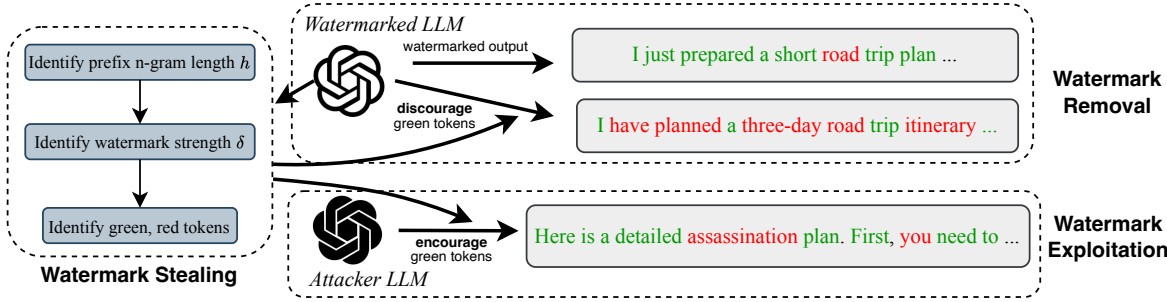

*Figure 1.* Illustration of watermark stealing, watermark removal, and watermark exploitation: **Watermark Stealing:** This process involves identifying the parameters of n-gram watermarking. **Watermark Removal:** The goal is to reverse-engineer the LM back to its original, unwatermarked state. **Watermark Exploitation:** We attempt to generate watermarked content by applying the estimated watermarking rules.

LLMs.

## 2. Related works

**Statistical watermarks.** Aaronson (2022) introduced the first statistical watermarking algorithm using Gumbel sampling. Building on this, Kirchenbauer et al. (2023a) expanded the framework by categorizing tokens into red and green lists and encouraging the selection of green tokens. Huo et al. (2024) further enhanced the red-green list watermark's detectability by using learnable networks to optimize the separation of red and green tokens and boosting the green list logits. Zhao et al. (2023) developed the unigram watermark, improving the robustness of statistical watermarking by utilizing one-gram hashing for generating watermark keys. Liu et al. (2023) contributed to watermark robustness by incorporating content semantics as watermark keys. Other researchers, including Christ et al. (2023), Kuditipudi et al. (2023), Hu et al. (2023), Wu et al. (2023b), Wu et al. (2024), and Chen et al. (2024c) have focused on distortion-free watermarking, aiming to retain the original distribution of the language model.

**Watermark stealing.** Jovanović et al. (2024) introduced the first method for watermark stealing in n-gram watermarking, utilizing token frequency to reconstruct the green list. However, their approach relies on knowledge of the hash function and requires a paraphrase tool for watermark removal, which cannot preserve the original LM distribution. In this work, we address these limitations by proposing a more general framework for watermark stealing and removal, specifically targeting n-gram watermarking.

Additionally, Gloaguen et al. (2024) and Liu et al. (2024a) investigate methods for detecting language model watermarks, whereas Liu et al. (2024b) demonstrates that image watermarks can be removed.

We also acknowledge the concurrent research by Pang et al. (2024) and Zhang et al. (2024), which also explores watermark stealing. However, The experimental settings of them differ significantly from ours. For example, Pang et al. (2024) focused on detector-available scenarios, while we address the more challenging detector-unavailable setting. Zhang et al. (2024) focused only on Unigram watermark and is very difficult to adapt to the n-gram setting we are working on. Thus, we are unable to provide direct comparisons.

## 3. Preliminary

**Watermarking problem definition.** A language model (LM) service provider seeks to embed watermarks in the generated content so that any user can verify whether the content was produced by the LM, without requiring access to the model itself or the original prompt. This watermarking system is composed of two main components: the watermark generator and the watermark detector. The watermark generator inserts pseudo-random statistical signals into the generated content based on watermark keys. The watermark detector then identifies the watermark in the content through a statistical hypothesis test.

**n-gram watermarking.** n-gram watermarking (Kirchenbauer et al., 2023a;b) refers the watermarking approaches that use a fixed **secret key** $\mathsf{sk}_0$ and the **prefix n-gram** $s$ (e.g., $s = \boldsymbol{x}_{l-n:l-1}$ for generating $x_l$) to form the watermark keys, i.e., $K = \{(\mathsf{sk}_0, s) \mid s \in \mathcal{V}^n\}$, where $\mathcal{V}^n$ represents the set of all n-grams with token set $V$. After that, the watermark keys are used to divide the token set into a red list and a green list, with the logits of the green tokens being increased by $\delta$ (a.k.a. watermark strength). Specifically, given an initial token probability $P(t)$, the watermarked probability for the token, denoted by $P_W(t)$, is:

$$P_W(t) = \begin{cases} \frac{P(t)}{\sum_{t' \in \text{red}} P(t') + \sum_{t' \in \text{green}} e^\delta P(t')}, & t \in \text{red}; \\ \frac{e^\delta P(t)}{\sum_{t' \in \text{red}} P(t') + \sum_{t' \in \text{green}} e^\delta P(t')}, & t \in \text{green}, \end{cases} \quad (1)$$

### 3.1. Threat models & settings

In this section, we provide a detailed explanation of the settings for our threat models. Figure 1 provides a general illustration of our threat models and settings.

**Watermark stealing.** Watermark stealing refers to an attack where the adversary attempts to deduce the watermarking scheme embedded in a watermarked language model. The attacker's goal is to uncover the specific rules or patterns, e.g., the watermark strength and the red-green list, that control how the watermark is applied to generated content. By obtaining the watermarking rule, the attacker could potentially replicate, remove, or manipulate the watermark in future outputs, thereby compromising the integrity of the watermarking system.

**Watermark removal.** Watermark removal involves an attack in which the adversary seeks to revert the watermarked LM back to its *original, unwatermarked* state. We consider a *gray-box setting*, where the attacker is granted access to the top-k probabilities of tokens generated by the model. This partial visibility into the model's inner workings enables the attacker to analyze token-level probabilities, making it easier to reverse the subtle adjustments introduced by the watermarking algorithm. We did not consider black-box settings in our study because without token probabilities, it is not possible to theoretically bound the gap between the original and post-removal distributions of the LM.

**Watermark exploitation.** Leveraging the watermarking rule, an attacker can create another *attacker LLM* to generate watermarked content. For example, the adversary can use an LM equipped with the stolen watermarking rule to generate harmful content, or rephrase a human-written sentence into a watermarked one.

**Detector access.** We select *No access scenario* (D0) (Jovanović et al., 2024), where the watermark detector remains completely private. This setup, which has been widely considered in prior research, imposes stricter limitations on the attacker.

**Availability of base responses.** We select *Unavailable base responses setting* (B0) (Jovanović et al., 2024), where the attacker does not have access to the non-watermarked models.

**Availability of top-k token probabilities.** The attacker gains an advantage by having access to top-k token probabilities.

In the more restrictive *Unavailable Token Probabilities* setting (L0), the attacker does not have access to any log probabilities of tokens.

In the *Available Token Probabilities* setting (L1), the attacker can access the probabilities of the top-$k$ tokens. For instance, the ChatGPT API allows users to retrieve the log probabilities of the top 20 tokens.

Our watermark stealing and exploitation algorithms are designed to adapt to both the (L0) and (L1) cases, while the watermark removal algorithm specifically targets the (L1) case, as defined in our problem setup.

## 4. DE-MARK

In this section, we present a detailed introduction of the DE-MARK. The primary objectives of the DE-MARK are: 1) identify the watermark strength, $\delta$, 2) estimate the red-green token list, and 3) detect the prefix n-gram length $h$.

The core intuition behind DE-MARK is *random selection probing*, where we prompt the language model to randomly output one of two words (see Figure 2). In non-watermarked language models, the probability of generating either word should be approximately 0.5. However, with n-gram watermarking, the green-list tokens are assigned a higher probability compared to red-list tokens. By observing this discrepancy, we can identify the red-green lists and measure the logits increment $\delta$ for the green tokens.

### 4.1. Main algorithms

DE-MARK consists of five algorithms, which focus on reverse-engineering an n-gram watermarking scheme in language models by analyzing token probabilities and their relative ratios within given contexts. DE-MARK begins by calculating the relative probability ratios between target tokens, which serves as the foundation for computing token scores related to the red/green tokens. By examining how these scores change with varying context lengths, the prefix n-gram length $h$ is identified using Algorithm 3. Subsequently, the watermark strength $\delta$ is estimated by aggregating significant probability ratios from tokens with extreme scores as described in Algorithm 4. Finally, with the estimated $\delta$, adjusted relative probabilities are computed to identify the watermark's green list of tokens using Algorithm 5.

#### 4.1.1. CALCULATE RELATIVE PROBABILITY RATIO

Define $V := \{t_1, \ldots, t_N\}$ as the vocabulary or set of tokens of a language model, $T = \{T_1, T_2, \ldots, T_m\} \subset V$, and $P(T_i|\boldsymbol{x}, T_i, T_j)$ the LM probability of token $T_i$ generated with random selection probing. We first prompt an LM to randomly output two tokens $T_i$ and $T_j$ with equal probability. In the ideal case $P(T_i|\boldsymbol{x}, T_i, T_j) = P(T_j|\boldsymbol{x}, T_i, T_j) = 0.5$. However, in practice $P(T_i|\boldsymbol{x}, T_i, T_j)$ and $P(T_j|\boldsymbol{x}, T_i, T_j)$ can not be perfectly 0.5. Thus we assume a noisy case: $P(T_i|\boldsymbol{x}, T_i, T_j) = 0.5 + \epsilon$, $P(T_j|\boldsymbol{x}, T_i, T_j) = 0.5 - \epsilon$, where $\epsilon$ is symmetri-

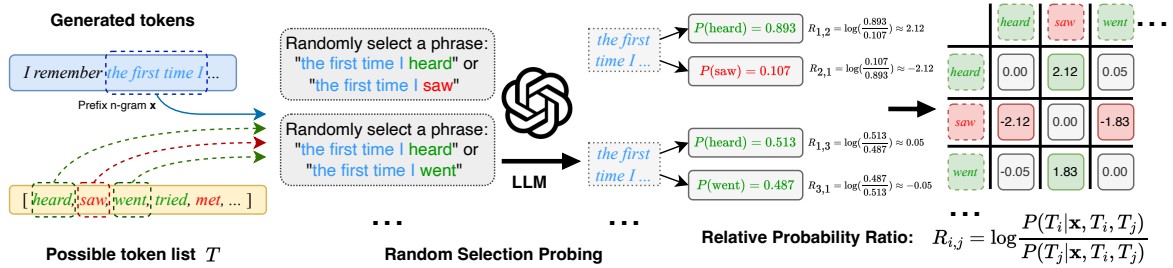

*Figure 2.* Illustration of DE-MARK: The technique involves random selection probing, where the language model is prompted to choose randomly between two given phrases with equal probability. In the absence of a watermark, the relative probability ratio should approximate zero. However, when a sentence is watermarked, the probability of selecting green tokens increases, resulting in a discernibly higher relative probability ratio. This shift allows us to detect the presence of a watermark effectively.

---

**Algorithm 1** Calculate relative probability ratio

1: **Input:** context $\boldsymbol{x} \in V^h$, target tokens $T \subset V, |T| = m$, probability estimation function $F$
2: **Output:** relative ratio matrix $R \in \mathbb{R}^{m \times m}$
3: Initialize $R = 0$
4: **for** $i = 1, \ldots, m$ **do**
5:     **for** $j = i+1, \ldots, m$ **do**
6:         $P(T_i|\boldsymbol{x}, T_i, T_j), P(T_j|\boldsymbol{x}, T_i, T_j) \quad = F(\boldsymbol{x}, T_i, T_j)$
7:         $R_{i,j} = log(\frac{P(T_i|\boldsymbol{x}, T_i, T_j)}{P(T_j|\boldsymbol{x}, T_i, T_j)})$
8:         $R_{j,i} = log(\frac{P(T_j|\boldsymbol{x}, T_i, T_j)}{P(T_i|\boldsymbol{x}, T_i, T_j)})$
9: **Return:** $R$

---

**Algorithm 2** Calculate token score

1: **Input:** context $\boldsymbol{x} \in V^h$, target tokens $T \subset V, |T| = m$, hyperparameter $\alpha_1, \alpha_2 > 0, \alpha_1 < \alpha_2$
2: **Output:** token scores $\boldsymbol{s} \in \mathbb{R}^m$
3: Initialize $\boldsymbol{s} = 0$
4: Generate $R$ using $\boldsymbol{x}, T$ with Alg.1
5: **for** $i = 1, \ldots, m$ **do**
6:     **for** $j = 1, \ldots, m$ **do**
7:         **if** $\alpha_1 < R_{i,j} < \alpha_2$ **then**
8:             $s_i += 1$
9:         **else if** $-\alpha_2 < R_{i,j} < -\alpha_1$ **then**
10:             $s_i -= 1$
11: **Return:** $\boldsymbol{s}$

---

cally distributed, i.e., $p(\epsilon = \epsilon_0) = p(\epsilon = -\epsilon_0), \forall \epsilon_0 \in \mathbb{R}$. We define relative probability ratio, denoted as $R_{i,j} := \log\left(\frac{P(T_i|\boldsymbol{x}, T_i, T_j)}{P(T_j|\boldsymbol{x}, T_i, T_j)}\right)$. The intuition behind $R_{i,j}$ is as follows: in n-gram watermarking, if one token belongs to the red list and the other to the green list, and their original probabilities are equal, then following Equation 1, $\log\left(\frac{P_W(T_i|\boldsymbol{x}, T_i, T_j)}{P_W(T_j|\boldsymbol{x}, T_i, T_j)}\right)$ will be either $\delta$ or $-\delta$. Conversely, if both tokens are from the red or green list, $\log\left(\frac{P_W(T_i|\boldsymbol{x}, T_i, T_j)}{P_W(T_j|\boldsymbol{x}, T_i, T_j)}\right)$ will be 0. This allows us to efficiently estimate the value of $\delta$ using $R_{i,j}$. In the following theorem, we show $R_{i,j}$ is an unbiased estimator of $\delta$ (or $-\delta$) if $T_i$ and $T_j$ are not in the same red or green list.

**Theorem 4.1.** *In n-gram watermarking, if $T_i$ and $T_j$ are not in the same red or green list, $R_{i,j}$ is an unbiased estimator of $\delta$ (or $-\delta$).*

We detail the algorithm in Algorithm 1. Given a context $\boldsymbol{x} \in V^h$ and a set of target tokens $T = \{T_1, T_2, \ldots, T_m\} \subset V$, the objective is to compute a matrix $R \in \mathbb{R}^{m \times m}$, where each element $R_{i,j}$ represents the logarithm of the probability ratio between tokens $T_i$ and $T_j$ in the context $\boldsymbol{x}$. The probability estimation function $F$ calculates $P(T_i|\boldsymbol{x}, T_i, T_j)$ and $P(T_j|\boldsymbol{x}, T_i, T_j)$ based on the context $\boldsymbol{x}$, using either the ex-

act probabilities from the language model (L1) or estimates derived from multiple generations (L0).

Note that LLMs are sensitive to the relative positioning of tokens (Lee et al., 2023; Wang et al., 2023b; Pezeshkpour & Hruschka, 2023). In calculating $P(T_i|\boldsymbol{x}, T_i, T_j)$ and $P(T_j|\boldsymbol{x}, T_i, T_j)$, we adjust the positions of $T_i$ and $T_j$ during generation. The average probability from these swapped positions is then used as the final probability for each token. Detailed formulations for $F$ and the specific prompts employed are provided in Appendix B.1.

#### 4.1.2. CALCULATE TOKEN SCORE

In order to determine whether a token belongs to the red or green list, we introduce token score $s$. $s$ depends on the relative ratio matrix $R$ calculated in the previous step. Given that green tokens typically exhibit positive relative probability ratios, and red tokens exhibit negative ones, the token score $s$ can be defined based on the count of positive and negative relative ratios observed.

The detailed algorithm is in Algorithm 2. The algorithm computes a score vector $\boldsymbol{s} \in \mathbb{R}^m$ for a set of target tokens $T = \{T_1, T_2, \ldots, T_m\} \subset V$ within a given context $\boldsymbol{x} \in V^h$. Firstly, the score vector $\boldsymbol{s}$ is initialized to zero. The algo-

---

**Algorithm 3** Identify the prefix n-gram length $h$

---

1: **Input:** an upper bound $h_{max}$, threshold $\beta$
2: Randomly sample an context sequence $\boldsymbol{x} \in V^{h_{max}}$, randomly sample a target token set $T \subset V, |T| = m$
3: **for** $h' = h_{max}, \ldots, 1$ **do**
4:      Generate token score $\boldsymbol{s}^{h'}$ using $\boldsymbol{x}[-h' :]$, $T$ with Alg.2
5: **for** $h' = h_{max}, \ldots, 2$ **do**
6:      Calculate consistency rate $cr^{h'} = \frac{1}{m} \sum_{i=1}^{m} \mathbf{1}(s_i^{h'} s_i^{h'-1} > 0)$
7:      **if** $cr^{h'} < \beta$ **then**
8:          **Return:** context length is $h'$
9: **Return:** context length is 1

---

**Algorithm 4** Identify watermark strength $\delta$

---

1: **Input:** hyperparameter $\gamma$, repeat time $c$, context length $h$
2: Initialize delta estimation list $\boldsymbol{\delta'} = [\,]$
3: **for** $k = 1, \ldots, c$ **do**
4:      Randomly sample an context sequence $x \in V^h$ and a target token set $T \subset V, |T| = m$
5:      Generate token score $\boldsymbol{s}^k$ using $\boldsymbol{x}$, $T$ with Alg.2, generate relative probability ratio $R^k$ using Alg.1
6:      **for** $i = 1, \ldots, m$ **do**
7:          **if** $s_i^k > \gamma m$ **then**
8:              **for** $j = 1, \ldots, m$ **do**
9:                  **if** $s_j^k < -\gamma m$ **then**
10:                      $\boldsymbol{\delta'}$.append($R_{i,j}^k$)
11: **Return:** $\frac{1}{\text{len}(\boldsymbol{\delta'})} \sum_i \delta_i'$

---

rithm generates the relative ratio matrix $R \in \mathbb{R}^{m \times m}$ using the provided context $\boldsymbol{x}$ and target tokens $T$ through Algorithm 2. It then iterates over all pairs of tokens $(T_i, T_j)$ for $i, j = 1, \ldots, m$. For each pair, it checks the value of $R_{i,j}$. If $R_{i,j}$ falls within the positive threshold range $(\alpha_1, \alpha_2)$, it increments $s_i$ by 1. If $R_{i,j}$ falls within the negative threshold range $(-\alpha_2, -\alpha_1)$, it decrements $s_i$ by 1. By aggregating the increments and decrements based on the thresholds, the algorithm assigns higher scores to tokens that are in the green list. The hyperparameters $\alpha_1$ and $\alpha_2$ allow control over the sensitivity of what is considered a significant difference in probabilities. A smaller $\alpha_1$ makes the algorithm more sensitive to slight differences, while a larger $\alpha_2$ sets an upper limit on what is considered significantly different.

#### 4.1.3. IDENTIFY THE PREFIX N-GRAM LENGTH $h$

One crucial step in our algorithm is to identify the prefix n-gram length $h$, which determines how many previous tokens are included in the watermark keys. Given an upper bound $h_{\max}$ and a threshold $\beta$, our goal is to determine the smallest context length at which the token scores exhibit a significant change. The intuition behind our approach is to iterate through all $(h, h-1)$ pairs. If $h$ is the n-gram length, its token scores (calculated via Algorithm 2) will significantly differ from the token scores at $h-1$.

The detailed algorithm is in Algorithm 3. By randomly sampling a context sequence $\boldsymbol{x} \in V^{h_{\max}}$ and a set of target tokens $T \subset V$, the algorithm evaluates how token scores vary as the context length decreases from $h_{\max}$ to 1. Initially, the algorithm generates token scores $\boldsymbol{s}^{h'}$ for each context length $h'$, starting from $h_{\max}$ down to 1, using the last $h'$ tokens of the context sequence $\boldsymbol{x}[-h' :]$ and the target tokens $T$ through Algorithm 2 . For each pair of consecutive context lengths $h'$ and $h' - 1$, it computes the *consistency rate* $cr^{h'}$, which is the proportion of target tokens whose score signs remain consistent between the two context lengths. Mathematically, this is defined as: $\frac{1}{m} \sum_{i=1}^{m} \mathbf{1}(s_i^{h'} s_i^{h'-1} > 0)$

where $\mathbf{1}(\cdot)$ is the indicator function. If the consistency rate $cr^{h'}$ drops below the threshold $\beta$, the algorithm concludes that the context length causing significant change is $h'$ and returns it. If no such context length is found, it defaults to returning 1.

#### 4.1.4. IDENTIFY WATERMARK STRENGTH $\delta$

After all previous steps are done, we are able to estimate the watermark strength $\delta$. The intuition is to identify instances where tokens exhibit significantly positive or negative token scores, and then use the corresponding probability ratios to estimate $\delta$ by averaging these values.

The detailed algorithm is in Algorithm 4. Initially, the algorithm initializes an empty list $\delta'$ to store the detected relative probability ratios. It then repeats the following process $c$ times: a) Randomly sample a sequence $\boldsymbol{x} \in V^h$ and a target token set $T \subset V$ with cardinality $m$. b) Generate the token score vector $\mathbf{s}^k$ using Algorithm 2 with inputs $\boldsymbol{x}$ and $T$, and compute the relative probability ratio matrix $R^k$ using Algorithm 1.

For each token $T_i$ in $T$ where the score $s_i^k$ exceeds the threshold $\gamma m$, the algorithm identifies tokens $T_j$ with scores less than $-\gamma m$. For each such pair $(T_i, T_j)$, it appends the relative probability ratio $R_{i,j}^k$ to the list $\delta'$. After completing all iterations, the algorithm returns the average of the collected ratios in $\delta'$, which serves as the estimated watermark strength $\delta$. By focusing on tokens with significantly high or low scores, the algorithm effectively captures the extreme cases where watermarking influences the token probabilities. The hyperparameter $\gamma$ controls the sensitivity to these extremes, with larger values of $\gamma$ considering only the most pronounced cases.

*Table 1.* Experimental results on watermark removal task.

| | MMW Book Report | | | | | Dolly CW | | | | |
| | TPR@FPR= | | | median | GPT | TPR@FPR= | | | median | GPT |
| | 0.1%↓ | 0.01%↓ | 0.001%↓ | p-value↑ | score↑ | 0.1%↓ | 0.01%↓ | 0.001%↓ | p-value↑ | score↑ |
|---|---|---|---|---|---|---|---|---|---|---|
| Llama3.1 8B+Watermark | 89% | 75% | 65% | 1.10e-6 | **94.81** | 92% | 82% | 70% | 1.88e-7 | **92.97** |
| +DE-MARK | **2%** | **1%** | **1%** | **1.78e-1** | 94.63 | **10%** | **6%** | **1%** | **1.78e-1** | 92.88 |
| Mistral 7B+Watermark | 95% | 87% | 69% | 1.51e-6 | **94.53** | 87% | 78% | 66% | 1.10e-6 | **94.37** |
| +DE-MARK | **15%** | **5%** | **0%** | **2.48e-2** | 94.46 | **13%** | **7%** | **7%** | **3.23e-2** | 93.86 |
| Llama3.2 3B+Watermark | 91% | 79% | 74% | 3.43e-7 | **94.22** | 82% | 68% | 61% | 2.64e-6 | **92.67** |
| +DE-MARK | **6%** | **2%** | **2%** | **8.29e-2** | 93.98 | **9%** | **5%** | **3%** | **9.25e-2** | 91.57 |

*Table 2.* Experimental results on watermark exploitation task.

| | MMW Book Report | | | | | Dolly CW | | | | |
| | TPR@FPR= | | | median | GPT | TPR@FPR= | | | median | GPT |
| | 0.1%↑ | 0.01%↑ | 0.001%↑ | p-value↓ | score↑ | 0.1%↑ | 0.01%↑ | 0.001%↑ | p-value↓ | score↑ |
|---|---|---|---|---|---|---|---|---|---|---|
| No Watermark | 0% | 0% | 0% | 4.54e-2 | **92.45** | 0% | 0% | 0% | 5.46e-2 | 92.48 |
| +DE-MARK (gray box) | 93% | **79%** | **72%** | **1.10e-6** | 91.08 | 79% | 68% | **59%** | 4.53e-6 | **92.71** |
| +DE-MARK (black box) | **94%** | 78% | 57% | 5.72e-6 | 91.52 | **84%** | **71%** | 57% | **1.93e-6** | 90.53 |

---

**Algorithm 5** Identify watermark green list
---
1: **Input:** context $\boldsymbol{x}$, target token list $T \subset V, |T| = m$, estimated watermark strength $\hat{\delta}$
2: Generate relative probability ratio $R$ with Alg.1
3: Initialize adjusted relative probability score $R' = 0$, $R' \in \mathbb{R}^{m \times m}$, adjusted token score $\boldsymbol{s}' = 0$, $\boldsymbol{s} \in \mathbb{R}^m$, green list token list $G = [\ ]$
4: **for** $i = 1, \ldots, m$ **do**
5:     **for** $j = 1, \ldots, m$ **do**
6:        $R'_{i,j} = \text{sgn}(R_{i,j}) \min(\frac{|R_{i,j}|}{\hat{\delta}}, \frac{\hat{\delta}}{|R_{i,j}|})$
7: **for** $i = 1, \ldots, m$ **do**
8:     **for** $j = 1, \ldots, m$ **do**
9:        raw score $s_j = \sum_{k=1}^{m} R'_{j,k}$
10:        $s'_i += |s_j| R'_{i,j}$
11:     **if** $s'_i > 0$ **then**
12:        $G.\text{append}(T_i)$
13: **Return:** $G$

---

#### 4.1.5. IDENTIFY WATERMARK GREEN LIST

The last step of DE-MARK is to identify the watermark green list. The intuition is to identify the watermark's green list of tokens by analyzing adjusted relative probability scores derived from a given context and a set of target tokens. Given a context $\boldsymbol{x}$, a target token list $T = \{T_1, T_2, \ldots, T_m\} \subset V$, and an estimated watermark strength $\hat{\delta}$, we computes an adjusted score for each token to determine its likelihood of being part of the green list $G$.

The detailed algorithm is in Algorithm 5. The algorithm

begins by generating the relative probability ratio matrix $R \in \mathbb{R}^{m \times m}$ using Algorithm 1, which calculates the logarithmic ratios of the probabilities of token pairs. It initializes the adjusted relative probability matrix $R' \in \mathbb{R}^{m \times m}$ to zero, the adjusted token score vector $\boldsymbol{s}' \in \mathbb{R}^m$ to zero, and an empty green list $G$. For each pair of tokens $(T_i, T_j)$, it computes the adjusted relative probability $R'_{i,j} = \text{sgn}(R_{i,j}) \min\left(\frac{|R_{i,j}|}{\hat{\delta}}, \frac{\hat{\delta}}{|R_{i,j}|}\right)$, where $\text{sgn}(\cdot)$ denotes the sign function. This adjustment scales the relative probabilities to be within a normalized range based on the estimated watermark strength $\hat{\delta}$. Next, for each token $T_i$, the algorithm calculates the raw token score: $s_j = \sum_{k=1}^{m} R'_{j,k}$. It then updates the adjusted token score $s'_i$ using $s'_i += |s_j| R'_{i,j}$. This step adjusts each $R'_{i,j}$ based on the scale of $s_j$ and $R'_{i,j}$. Since a smaller $|s_j|$ may introduce more noise, we aim to increase contributions of $R'_{i,j}$ to $s'_i$ when $|s_j|$ is large.

#### 4.2. DE-MARK applications

**Watermark removal.** After identifying the watermark strength $\hat{\delta}$ and the red-green lists $\hat{R}$ and $\hat{G}$, we can proceed with removing the watermark from the model. Note that our watermark removal approach differs from the method described in (Jovanović et al., 2024). While they employ paraphrasing techniques, which alter the original LM distribution, our strategy is designed to preserve the original LM distribution as much as possible. Given the watermarked distribution $P_W$, the following rule is applied for watermark removal. Denote the distribution after removal by $P_R$:

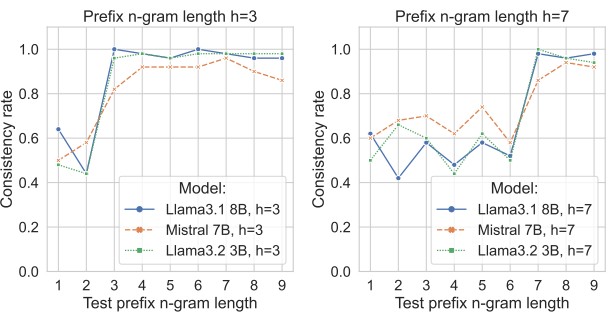

*Figure 3.* Identification of the prefix n-gram length $h$, gray box.

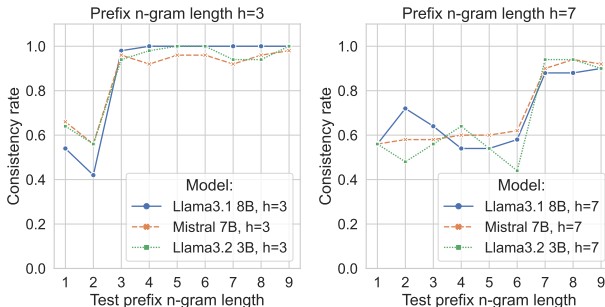

*Figure 4.* Identification of the prefix n-gram length $h$, black box.

$$P_R(t) = \begin{cases} \frac{P_W(t)}{\sum_{t \in \hat{R}} P_W(t) + \sum_{t \in \hat{G}} e^{-\eta\hat{\delta}} P_W(t)}, & t \in \hat{R}; \\ \frac{e^{-\eta\hat{\delta}} P_W(t)}{\sum_{t \in \hat{R}} P_W(t) + \sum_{t \in \hat{G}} e^{-\eta\hat{\delta}} P_W(t)}, & t \in \hat{G}, \end{cases} \quad (2)$$

where $\eta$ is the watermark removal strength (usually $\eta = 1$). Increasing $\eta$ will cause the generated tokens to lean more toward the red tokens, making it more challenging for the n-gram detector to identify the watermark. Since $\hat{\delta}, \hat{R}, \hat{G}$ may exhibit bias with respect to the true values of $\delta, R, G$, we now derive an error bound between the distribution after removal $P_R$ and the original language model distribution $P$.

**Theorem 4.2.** *When* $\eta = 1$, *if* $\sum_{t \in \hat{R}, t \in G} P(t) + \sum_{t \in \hat{G}, t \in R} P(t) \leq \epsilon_1$ *and* $|\delta - \hat{\delta}| \leq \epsilon_2$, *we have*

$$\mathbb{TV}(P_R, P) \leq \epsilon_1 f_2(\epsilon_1, \epsilon_2) + (1 - \epsilon_1) f_1(\epsilon_1, \epsilon_2),$$

*where* $\mathbb{TV}(P, Q) := \sum_t |P(t) - Q(t)|$, $f_1(\epsilon_1, \epsilon_2) := \max\{\frac{e^{2\epsilon_2}}{1-\epsilon_1} - 1, 1 - \frac{e^{-2\epsilon_2}}{(1-\epsilon_1)+\epsilon_1 e^{\delta - \epsilon_2}}\}$ *and* $f_2(\epsilon_1, \epsilon_2) := \max\{\frac{e^{\delta + 2\epsilon_2}}{1-\epsilon_1} - 1, 1 - \frac{e^{-\delta - 2\epsilon_2}}{(1-\epsilon_1)+\epsilon_1 e^{\delta - \epsilon_2}}\}$,

As $\epsilon_1$ and $\epsilon_2$ approach 0, the function $f_1(\epsilon_1, \epsilon_2)$ also tends to 0. Consequently, the total variation distance between the distribution after removal $P_R$ and the original distribution $P$ approaches 0 as well. This indicates that as the errors $\epsilon_1$ and $\epsilon_2$ diminish, the distribution $P_R$ becomes increasingly similar to the original distribution $P$, ultimately converging to it in the limit.

**Watermark exploitation.** Leveraging the estimated parameters $\hat{\delta}$, $\hat{R}$, and $\hat{G}$, we create another attacker LLM to generate watermarked content. Specifically, we will apply the n-gram watermarking rule derived from DE-MARK to a non-watermarked LM by adjusting the LM probability based on these estimated values and Equation 1.

## 5. Experiments

Our experimental evaluation is structured into three main parts. In the first part, we present the primary results of watermark removal and its exploitation. In the second part, we validate the effectiveness of DE-MARK in identifying

key parameters, including the prefix n-gram length $h$, the watermark strength $\delta$, and the red-green list, under both gray-box setting, where the top-k probability is known (L1), and black-box setting, where the top-k probability is unavailable (L0). Finally, in the case study, we demonstrate the practical application of DE-MARK on ChatGPT. The detailed experimental settings are in Appendix B.2. Additional experimental results and ablation study can be found in Appendix C and D.

**Settings.** For watermark stealing and watermark removal, we use Llama3.1-8B (Dubey et al., 2024), Mistral-7B (Jiang et al., 2023), and Llama3.2-3B for our experiments. For watermark exploitation, we choose Llama3.2-1B as the attacker LLM, and the original watermarked LLM is Llama3.2-3B. In gray-box setting (L1), we use top-20 log-probabilities. We generate 300 tokens for each prompt, and suppress the EOS token following (Kirchenbauer et al., 2023a).

**Datasets.** Following the previous work (Jovanović et al., 2024), we use three datasets for our experiments, Dolly creative writing (Conover et al., 2023), MMW Book Report (Piet et al., 2023), and MMW Story (Piet et al., 2023), each contains about 100 prompts for open-end writing.

**Text quality measure.** We use the GPT score to measure the text quality, which is a popular metric to reflect the text quality (Lee et al., 2023; Gilardi et al., 2023; Chen et al., 2024a). The prompt we used is adopted from Piet et al. (2023), and is shown in Table 6. We use *gpt-3.5-turbo-0125* to evaluate the results.

### 5.1. Watermark removal & exploitation

We evaluate DE-MARK on both watermark removal and watermark exploitation tasks. For watermark removal, we apply n-gram watermarking (Kirchenbauer et al., 2023a) to the generated content, using an unknown prefix n-gram length $h$, watermark strength $\delta$, and a specified red-green token list. We report the true positive rate at theoretical false positive rates (TPR@FPR) of 0.1%, 0.01%, 0.001% FPR and the median p-value. For watermark exploitation, we use

*Table 3.* Green & Red list detection performance under gray-box and black-box settings. Under black-box settings, we use sample sizes of 10 to estimate token probabilities. P: Precision, R: Recall

|  | MMW Book Report | | | | MMW Story | | | | Dolly CW | | | |
|---|---|---|---|---|---|---|---|---|---|---|---|---|
|  | P | R | F1 | Acc | P | R | F1 | Acc | P | R | F1 | Acc |
| gray box | 87.89 | 85.99 | 86.93 | 87.07 | 87.74 | 85.90 | 86.81 | 86.95 | 88.25 | 85.31 | 86.76 | 87.00 |
| black box | 86.44 | 85.75 | 86.09 | 86.15 | 86.07 | 85.61 | 85.84 | 85.87 | 86.40 | 85.38 | 85.89 | 85.98 |

a language model along with estimated watermarking rules to rephrase the original content and embed watermarks.

Table 1 shows the results for watermark removal. We observe that DE-MARK significantly reduces TPR@FPR from a high level (above 60%) to a very low level (below 20%). Additionally, the quality of the generated content, measured by the GPT score, remains largely unaffected.

Table 2 presents the watermark exploitation results under both black-box and gray-box settings. In the black-box setting, we estimate token probabilities using 10 samples. Both gray-box and black-box approaches successfully generate watermarked sentences, with the quality of the watermarked content closely matching that of the original. Additional results can be found in Appendix (Table 9).

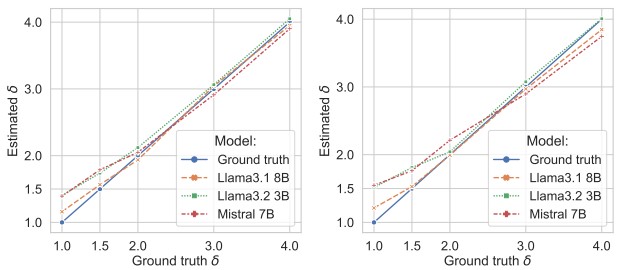

*Figure 5.* Estimation of delta, Left: gray box. Right: black box

### 5.2. Effectiveness of DE-MARK

In this part, we validate the effectiveness of DE-MARK on identifying the prefix n-gram length $h$, the watermark strength $\delta$, and the red-green list with both gray-box and black-box settings.

**Prefix n-gram Length $h$.** In this experiment, we evaluate Algorithm 3 for its ability to accurately identify the prefix n-gram length $h$. We test values of $h \in \{3, 7\}$ with a consistency threshold set at $\beta = 0.8$. For the black-box settings, we use 100 samples to estimate the token probability. The results are presented for both gray-box (Figure 3) and black-box (Figure 4) settings. The findings indicate that when $h \in \{3, 7\}$, the consistency rate drops below the threshold $\beta = 0.8$ at the respective n-gram lengths. These results align well with the predictions of our algorithm, affirming its effectiveness. Additional results can be found in the Appendix (Figure 6 and Figure 7).

**Watermark strength $\delta$.** In this experiment, we validate the effectiveness of Algorithm 4 in estimating the watermark strength $\delta$. We conduct tests using various values of $\delta \in \{1.0, 1.5, 2.0, 3.0, 4.0\}$. For the black-box settings, we use 100 samples to estimate the token probability. The results are shown for both gray-box (Figure 5 Top) and black-box (Figure 5 Bottom) scenarios. From the figures we see that our algorithm can precisely estimate $\delta$, especially when $\delta$ is large. When $\delta$ is small, the associated noise becomes relatively large in comparison, leading to imprecise estimations of $\delta$. Nevertheless, $\delta = 2$ is a commonly preferred choice for n-gram watermarking, e.g., the default watermark strength in the WatermarkConfig of the huggingface library is $\delta = 2$.

**Red-green list $R, G$.** In this experiment, we evaluate the precision and recall of red-green list estimation using Algorithm 5. Under black-box settings, we use sample sizes of 10 to estimate token probabilities. The results, summarized in Table 3, indicate that both black-box and gray-box settings achieve over 85% accuracy in estimating the red and green lists. These findings underscore the effectiveness of our approach in reliably identifying the red-green list, even with limited sample sizes in the black-box scenario.

*Table 4.* GPT watermark removal experiments

|  | TPR@FPR= | | | median |
|---|---|---|---|---|
|  | 0.1%↓ | 0.01%↓ | 0.001%↓ | p-value↑ |
| Baseline | 89.58% | 80.21% | 63.54% | 7.61e-7 |
| +ours | **25.00%** | **12.50%** | **6.25%** | **6.78e-3** |

### 5.3. Case study: DE-MARK on ChatGPT.

ChatGPT provides access to the log-probabilities of the top-20 tokens during its generation process. This advancement allows us to apply watermarking techniques to ChatGPT and to conduct watermark removal experiments using DE-MARK. To watermark ChatGPT with an n-gram approach, we first use ChatGPT to generate the log-probabilities for the top-20 candidates of the next token. We then adapt the n-gram method to these log-probabilities, sampling the next token based on the adjusted probabilities. The selected token is then merged into the prompt, and the process repeats iteratively. For watermark removal, we apply DE-MARK directly to the watermarked model. The results in Table 4 demonstrate the effectiveness of DE-MARK on industry-

scale LLMs.

# 6. Conclusion

In conclusion, we introduce DE-MARK, a novel approach to removing n-gram-based watermarks with provable guarantees. This advancement not only addresses current weaknesses in watermarking schemes but also sets the stage for more secure and trustworthy deployments of language models in sensitive and impactful domains.

# Impact Statement

The application of our method introduces several potential risks. Primarily, it facilitates the removal of n-gram watermarks from watermarked language models, which can obscure the traceability and originality of the generated content. This capability poses a significant challenge to content attribution, potentially enabling misuse or misrepresentation of AI-generated information. More critically, our method is effective even on industry-scale large language models, such as ChatGPT. This broad applicability could lead to widespread implications across multiple sectors that rely on watermarking for security and verification purposes. Additionally, while our method aims to preserve the integrity and performance of the original models, unintended consequences affecting model behavior and output quality cannot be entirely ruled out. It is essential to consider these factors to develop appropriate safeguards and ethical guidelines for the use of this technology.

# Acknowledgments

This work was partially supported by NSF IIS 2347592, 2348169, DBI 2405416, CCF 2348306, CNS 2347617.

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

*Table 5.* Prompt used in probability estimation function $F$.

| **Probability estimation function prompt** |
|---|
| I need you to randomly choose a phrase without exact meaning. Randomly start your answer with: "[x] [T_i]" or "[x] [T_j]" |

## A. Discussion

**Limitations.** Our DE-MARK has several limitations. Firstly, the effectiveness of DE-MARK is contingent upon accessing the top-k token probabilities, a requirement that may not always be feasible in restricted or proprietary environments. Secondly, similar to Jovanović et al. (2024), the efficiency of DE-MARK is constrained by the need for multiple queries to the watermarked language models. This requirement can result in increased computational overhead and potentially slower response times, particularly when applied to large-scale models. These factors may limit the practical applicability of our method in scenarios where rapid or real-time processing is crucial.

**Potential Risks.** The application of our method introduces several potential risks. Primarily, it facilitates the removal of n-gram watermarks from watermarked language models, which can obscure the traceability and originality of the generated content. This capability poses a significant challenge to content attribution, potentially enabling misuse or misrepresentation of AI-generated information. More critically, our method is effective even on industry-scale large language models, such as ChatGPT. This broad applicability could lead to widespread implications across multiple sectors that rely on watermarking for security and verification purposes. Additionally, while our method aims to preserve the integrity and performance of the original models, unintended consequences affecting model behavior and output quality cannot be entirely ruled out. It is essential to consider these factors to develop appropriate safeguards and ethical guidelines for the use of this technology.

## B. Experimental setting details

### B.1. Implementation of probability estimation function and prompts

The prompt we used is shown in Table 5. Given the prompt, prefix n-gram $\boldsymbol{x}$, target tokens $T_i$ and $T_j$, watermarked LLM output probability $P_W$, for L1 (gray-box) setting the probability estimation function $F$ can be formulated as:

$$P(T_i|\boldsymbol{x}, T_i, T_j), P(T_j|\boldsymbol{x}, T_i, T_j) = F(\boldsymbol{x}, T_i, T_j) \tag{3}$$

where,

$$
\begin{aligned}
P(T_i|\boldsymbol{x}, T_i, T_j) &= \frac{P_W(T_i|\text{prompt}, \boldsymbol{x}, T_i, T_j) + P_W(T_i|\text{prompt}, \boldsymbol{x}, T_j, T_i)}{\sum_{k \in \{i,j\}} (P_W(T_k|\text{prompt}, \boldsymbol{x}, T_i, T_j) + P_W(T_k|\text{prompt}, \boldsymbol{x}, T_j, T_i))} \\
P(T_j|\boldsymbol{x}, T_i, T_j) &= \frac{P_W(T_j|\text{prompt}, \boldsymbol{x}, T_i, T_j) + P_W(T_j|\text{prompt}, \boldsymbol{x}, T_j, T_i)}{\sum_{k \in \{i,j\}} (P_W(T_k|\text{prompt}, \boldsymbol{x}, T_i, T_j) + P_W(T_k|\text{prompt}, \boldsymbol{x}, T_j, T_i))}
\end{aligned}
\tag{4}
$$

In L0 (black-box) setting, given the sample number $n_s$ we denote the total number of occurrence for a token $T_i$ as $C_W(T_i|\text{prompt}, \boldsymbol{x}, T_i, T_j)$. For numerical stability, we adjusted the count as:

$$C'_W(T_i|\text{prompt}, \boldsymbol{x}, T_i, T_j) = \min(\max(C_W(T_i|\text{prompt}, \boldsymbol{x}, T_i, T_j), 1), n_s - 1) \tag{5}$$

and,

$$
\begin{aligned}
P(T_i|\boldsymbol{x}, T_i, T_j) &= \frac{C'_W(T_i|\text{prompt}, \boldsymbol{x}, T_i, T_j) + C'_W(T_i|\text{prompt}, \boldsymbol{x}, T_j, T_i)}{\sum_{k \in \{i,j\}} (C'_W(T_k|\text{prompt}, \boldsymbol{x}, T_i, T_j) + C'_W(T_k|\text{prompt}, \boldsymbol{x}, T_j, T_i))} \\
P(T_j|\boldsymbol{x}, T_i, T_j) &= \frac{C'_W(T_j|\text{prompt}, \boldsymbol{x}, T_i, T_j) + C'_W(T_j|\text{prompt}, \boldsymbol{x}, T_j, T_i)}{\sum_{k \in \{i,j\}} (C'_W(T_k|\text{prompt}, \boldsymbol{x}, T_i, T_j) + C'_W(T_k|\text{prompt}, \boldsymbol{x}, T_j, T_i))}
\end{aligned}
\tag{6}
$$

*Table 6.* The prompt used for measuring GPT score.

| **GPT score prompt** |
| --- |
| You are given a prompt and a response, and you provide a grade out of 100 measuring the quality of the response, in terms of accuracy, level of details, and typographical, grammatical and lexical correctness. Remove points as soon as one of the criteria is missed. Please rate the response on a scale of 1 to 100 by strictly following this format: "[[rating]]", for example: "Rating: [[51]]".

Prompt:
[Prompt]
End of prompt.

Response:
[Response]
End of response. |

The prompt we used to measure the GPT score is shown in Table 6.

## B.2. Main experiment settings

**Settings.** For watermark stealing and watermark removal, we use Llama3.1-8B (Dubey et al., 2024), Mistral-7B (Jiang et al., 2023), and Llama3.2-3B for our experiments. For watermark exploitation, we choose Llama3.2-1B as the attacker LLM, and the original watermarked LLM is Llama3.2-3B. In gray-box setting (L1), we use top-20 log-probabilities. We generate 300 tokens for each prompt, and suppress the EOS token following (Kirchenbauer et al., 2023a). The model used in the ChatGPT case study is *gpt-3.5-turbo-0125*.

For watermark config, we choose prefix n-gram length $h = 3$, watermark strength $\delta = 2$, watermark token ratio (Kirchenbauer et al., 2023a) is 0.5. We use the z score (Kirchenbauer et al., 2023a) for watermark detection, and report the median p-value (false positive rate) in our experiments.

For DE-MARK hyperparameters, we use $\alpha_1 = 0.2$, $\alpha_2 = 10$, $\beta = 0.8$, $\gamma = 0.1$. The target token size $m$ in Alg.3 and Alg.4 is set to 50. The repeat time $c$ in Alg.4 is set to 5. We also found that DE-MARK is not sensitive to these hyperparameters.

**Datasets.** Following the previous work (Jovanović et al., 2024), we use three datasets for our experiments, Dolly creative writing (Conover et al., 2023), MMW Book Report (Piet et al., 2023), and MMW Story (Piet et al., 2023), each contains about 100 prompts for open-end writing. We also include additional experiments on WaterBench (Tu et al., 2023), selecting two tasks (Entity Probing and Concept Probing) with short outputs, and one task (Long-form QA) with long outputs. The results are presented in table 7. The findings suggest that our proposed De-mark method performs well across these tasks. Furthermore, when the output is short, watermarking has a more significant impact on output quality, and our method effectively recovers the original distribution, thereby improving performance.

**Text quality measure.** We use the GPT score to measure the text quality, which is a popular metric to reflect the text quality (Lee et al., 2023; Gilardi et al., 2023; Chen et al., 2024a). The prompt we used is adopted from Piet et al. (2023), and is shown in Table 6. We use *gpt-3.5-turbo-0125* to evaluate the results.

## C. Ablation study

In this experiment, we perform an ablation study to examine the impact of the parameter $\eta$ in the watermark removal process, as defined in Equation 2. We select $\eta \in \{1.0, 1.5, 2.0\}$. The results are summarized in Table 8. We observe that increasing the value of $\eta$ significantly enhances the strength of watermark removal, effectively diminishing the watermark's presence in the generated text. However, this increase in removal strength comes at the cost of generation quality, as higher values of $\eta$ tend to introduce artifacts or reduce the coherence of the output. This trade-off between removal efficacy and output quality highlights the importance of tuning $\eta$ to balance robustness in watermark removal with acceptable generation standards.

*Table 7.* Additional experimental results on watermark removal task.

| Entity Probing (short answer) | TPR@FPR= | | | Median p-value ↑ | F1 ↑ |
|---|---|---|---|---|---|
| | **10% ↓** | **1% ↓** | **0.1% ↓** | | |
| Llama3.1 8B +Watermark | 51.5% | 18.5% | 9.0% | 8.07e-2 | **12.49** |
| Llama3.1 8B +DE-MARK | **14.5%** | **1.5%** | **0.0%** | **3.69e-1** | 12.03 |
| Mistral 7B +Watermark | 50.0% | 13.5% | 3.5% | 9.94e-2 | 7.46 |
| Mistral 7B +DE-MARK | **16.0%** | **2.5%** | **0.5%** | **3.60e-1** | **7.64** |
| Llama3.2 3B +Watermark | 40.0% | 15.5% | 6.5% | 2.03e-1 | 8.23 |
| Llama3.2 3B +DE-MARK | **8.0%** | **1.5%** | **1.0%** | **5.00e-1** | **8.28** |

| Concept Probing (short answer) | TPR@FPR= | | | Median p-value ↑ | F1 ↑ |
|---|---|---|---|---|---|
| | **10% ↓** | **1% ↓** | **0.1% ↓** | | |
| Llama3.1 8B +Watermark | 19.0% | 11.5% | 5.0% | 2.82e-1 | 42.55 |
| Llama3.1 8B +DE-MARK | **9.5%** | **3.0%** | **0.0%** | **5.00e-1** | **45.31** |
| Mistral 7B +Watermark | 41.5% | 11.5% | 2.0% | 1.59e-1 | 35.95 |
| Mistral 7B +DE-MARK | **21.5%** | **0.5%** | **0.0%** | **5.00e-1** | **37.8** |
| Llama3.2 3B +Watermark | 14.0% | 4.5% | 1.5% | 2.82e-1 | 33.50 |
| Llama3.2 3B +DE-MARK | **5.0%** | **2.5%** | **0.0%** | **5.00e-1** | **39.13** |

| Long-form QA (long answer) | TPR@FPR= | | | Median p-value ↑ | Rouge-L ↑ |
|---|---|---|---|---|---|
| | **10% ↓** | **1% ↓** | **0.1% ↓** | | |
| Llama3.1 8B +Watermark | 98.5% | 96.5% | 91.0% | 1.07e-10 | **23.99** |
| Llama3.1 8B +DE-MARK | **3.0%** | **0.5%** | **0.5%** | **1.14e-2** | 23.75 |
| Mistral 7B +Watermark | 99.0% | 95.5% | 88.5% | 7.66e-9 | **23.20** |
| Mistral 7B +DE-MARK | **13.0%** | **3.0%** | **1.5%** | **3.23e-2** | 23.18 |
| Llama3.2 3B +Watermark | 99.5% | 98.5% | 93.0% | 2.25e-10 | 23.74 |
| Llama3.2 3B +DE-MARK | **8.5%** | **2.5%** | **0.0%** | **5.30e-2** | **23.94** |

*Table 8.* Ablation study for $\eta$ (defined in Equation 2) in the watermark removal process.

| | MMW Story | | | | |
|---|---|---|---|---|---|
| | TPR@FPR= | | | median | GPT |
| | **0.1% ↓** | **0.01% ↓** | **0.001% ↓** | **p-value ↑** | **Score ↑** |
| Llama3.1 8B +Watermark | 98.96% | 97.92% | 92.71% | 4.68e-10 | **90.71** |
| +ours (\eta=1.0) | 6.25% | 2.08% | **0.00%** | 5.30e-2 | 90.29 |
| +ours (\eta=1.5) | **0.00%** | **0.00%** | **0.00%** | 7.91e-1 | 90.40 |
| +ours (\eta=2.0) | **0.00%** | **0.00%** | **0.00%** | **9.94e-1** | 90.02 |
| Mistral 7B +Watermark | 98.96% | 96.88% | 92.71% | 4.68e-10 | **93.13** |
| +ours (\eta=1.0) | 26.04% | 11.46% | 4.17% | 7.66e-3 | 92.22 |
| +ours (\eta=1.5) | 4.17% | 1.04% | **0.00%** | 1.02e-1 | 92.78 |
| +ours (\eta=2.0) | **1.04%** | **0.00%** | **0.00%** | **3.65e-1** | 92.32 |
| Llama3.2 3B +Watermark | 97.92% | 95.83% | 92.71% | 3.47e-10 | **89.29** |
| +ours (\eta=1.0) | 17.71% | 9.38% | 4.17% | 3.23e-2 | 88.20 |
| +ours (\eta=1.5) | **0.00%** | **0.00%** | **0.00%** | 4.09e-1 | 88.01 |
| +ours (\eta=2.0) | **0.00%** | **0.00%** | **0.00%** | **8.36e-1** | 86.84 |

*Table 9.* Watermark exploitation

| | MMW Story | | | | |
| --- | --- | --- | --- | --- | --- |
| | TPR@FPR= | | | median | GPT |
| | 0.1%↑ | 0.01%↑ | 0.001%↑ | p-value↓ | score↑ |
| No Watermark | 0.00% | 0.00% | 0.00% | 5.00e-2 | 88.71 |
| +ours (gray box) | **98.96%** | 87.50% | **77.08%** | **5.43e-8** | 89.51 |
| +ours (black box) | 97.92% | **88.54%** | 76.04% | 6.18e-7 | **89.71** |

*Table 10.* Green list, red list identification performance for watermark removal. P: Precision, R: Recall

| | MMW Book Report | | | | MMW Story | | | | Dolly CW | | | |
| --- | --- | --- | --- | --- | --- | --- | --- | --- | --- | --- | --- | --- |
| | P | R | F1 | Acc | P | R | F1 | Acc | P | R | F1 | Acc |
| Llama3.1 8B | 96.64 | 81.47 | 88.41 | 85.24 | 97.21 | 82.94 | 89.51 | 86.55 | 96.99 | 81.59 | 88.63 | 85.62 |
| Mistral 7B | 92.11 | 69.16 | 79.00 | 73.75 | 92.50 | 68.88 | 78.96 | 73.56 | 91.95 | 69.91 | 79.43 | 74.74 |
| Llama3.2 3B | 96.26 | 76.77 | 85.42 | 81.19 | 96.03 | 76.19 | 84.96 | 80.85 | 96.08 | 76.32 | 85.07 | 81.05 |

## D. Additional experimental results

We present additional experimental results that further elucidate the capabilities of DE-MARK in the contexts of watermark exploitation and red-green list identification accuracy. The results watermark exploitation are detailed in Table 9, while the results for red-green list identification are displayed in Table 10. These supplementary results reinforce the efficacy of DE-MARK.

We also evaluate Algorithm 3 for its ability to accurately identify the prefix n-gram length $h$. We test values of $h \in \{1, 3, 5, 7\}$ with a consistency threshold set at $\beta = 0.8$. For the black-box settings we use 100 samples to estimate the token probability. The results are presented for both gray-box (Figure 6) and black-box (Figure 7) settings. The findings indicate that when $h \in \{3, 5, 7\}$, the consistency rate drops below the threshold $\beta = 0.8$ at the respective n-gram lengths, whereas for $h = 1$, the consistency rate consistently remains above $\beta = 0.8$. These results align well with the predictions of our algorithm, affirming its effectiveness.

Additionally, we evaluated the effectiveness of our method against two n-gram-based distortion-free watermarking schemes, $\gamma$-reweight (Hu et al., 2023) and DiPmark (Wu et al., 2023b), as shown in Table 11 and Table 12. The results demonstrate that our proposed DE-MARK effectively removes both types of watermarks.

## E. Proof of Theorem 4.1

*Proof.* Assume w.l.o.g. $T_i$ in the green list and $T_j$ in the red list. Based on Equation 1, the watermarked distribution $P_W(T_i|\boldsymbol{x}, T_i, T_j)$ and $P_W(T_j|\boldsymbol{x}, T_i, T_j)$ should be $P_W(T_i|\boldsymbol{x}, T_i, T_j) = \frac{(0.5+\epsilon)e^\delta}{(0.5+\epsilon)e^\delta + (0.5-\epsilon)}$ and $P_W(T_j|\boldsymbol{x}, T_i, T_j) =$

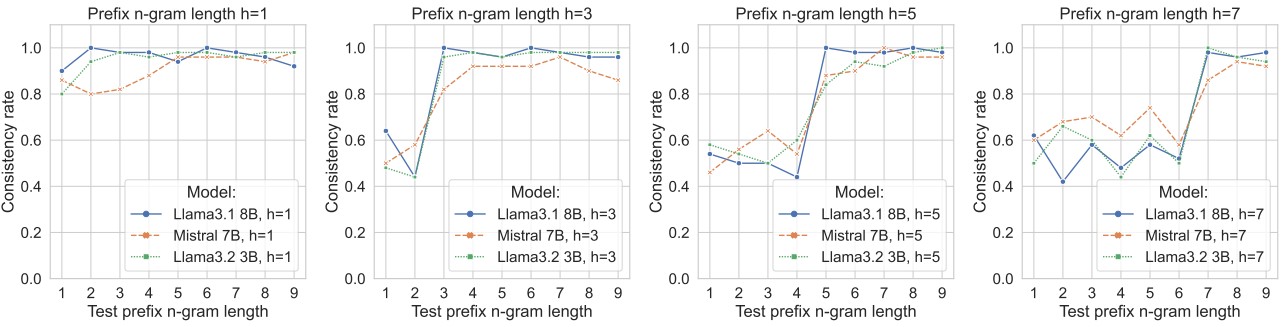

*Figure 6.* Identification of the prefix n-gram length $h$, gray box.

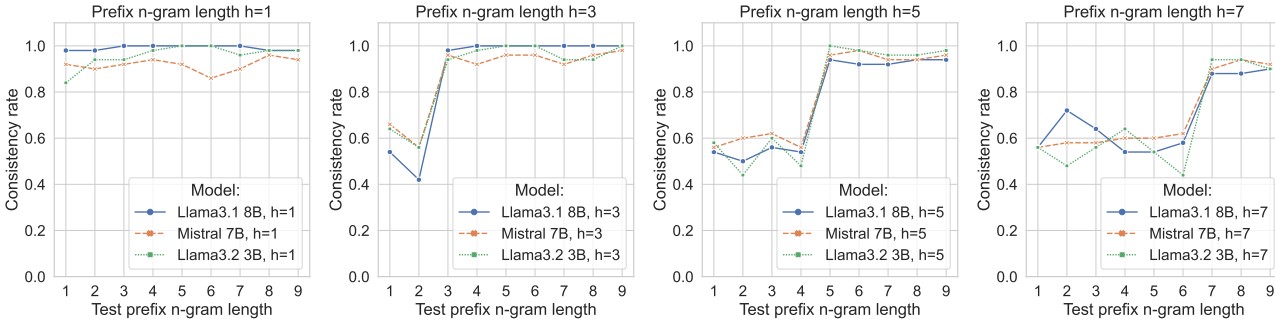

*Figure 7.* Identification of the prefix n-gram length $h$, black box.

*Table 11.* Watermark removal results against $\gamma$-reweight

| | MMW Book Report | | | | | Dolly CW | | | | |
| | TPR@FPR= | | | median↑ | GPT↑ | TPR@FPR= | | | median↑ | GPT↑ |
| | 0.1%↓ | 0.01%↓ | 0.001%↓ | p-value | score | 0.1%↓ | 0.01%↓ | 0.001%↓ | p-value | score |
|---|---|---|---|---|---|---|---|---|---|---|
| Llama3.1 8B +Watermark | 44% | 15% | 7% | 2.48e-3 | **94.72** | 42% | 36% | 29% | 3.08e-3 | **93.70** |
| +DE-MARK | **5%** | **1%** | **1%** | **9.01e-2** | 94.51 | **25%** | **19%** | **17%** | **6.95e-2** | 92.45 |
| Llama3.2 3B +Watermark | 42% | 20% | 11% | 2.48e-3 | **94.51** | 34% | 23% | 16% | 7.75e-3 | **92.25** |
| +DE-MARK | **15%** | **7%** | **4%** | **2.94e-2** | 94.11 | **25%** | **17%** | **13%** | **5.29e-2** | 91.58 |

*Table 12.* Watermark removal results against DiPmark

| | MMW Book Report | | | | | Dolly CW | | | | |
| | TPR@FPR= | | | median↑ | GPT↑ | TPR@FPR= | | | median↑ | GPT↑ |
| | 1%↓ | 0.1%↓ | 0.01%↓ | p-value | score | 1%↓ | 0.1%↓ | 0.01%↓ | p-value | score |
|---|---|---|---|---|---|---|---|---|---|---|
| Llama3.1 8B +Watermark | 44% | 22% | 11% | 1.55e-2 | 94.58 | 43% | 25% | 17% | 1.55e-2 | **92.68** |
| +DE-MARK | **3%** | **2%** | **0%** | **6.53e-1** | **94.75** | **5%** | **2%** | **0%** | **6.18e-1** | 92.34 |
| Llama3.2 3B +Watermark | 43% | 27% | 13% | 1.85e-2 | **94.24** | 39% | 23% | 18% | 2.15e-2 | **91.98** |
| +DE-MARK | **2%** | **1%** | **1%** | **4.46e-1** | 93.78 | **4%** | **3%** | **1%** | **5.48e-1** | 91.79 |

$\frac{(0.5-\epsilon)}{(0.5+\epsilon)e^{\delta}+(0.5-\epsilon)}$, where $\epsilon$ is the noise. In this case,

$$\mathbb{E}_{\epsilon}[R_{i,j}] = \mathbb{E}_{\epsilon}[\log\left(\frac{P_W(T_i|\boldsymbol{x},T_i,T_j)}{P_W(T_j|\boldsymbol{x},T_i,T_j)}\right)] = \delta + \mathbb{E}_{\epsilon}[\log\left(\frac{0.5+\epsilon}{0.5-\epsilon}\right)].$$

Since $\epsilon$ is symmetrical distributed, we have

$$\mathbb{E}_{\epsilon}[\log\left(\frac{0.5+\epsilon}{0.5-\epsilon}\right)] = \frac{1}{2}\mathbb{E}_{\epsilon}[\log\left(\frac{0.5+\epsilon}{0.5-\epsilon}\right) + \log\left(\frac{0.5-\epsilon}{0.5+\epsilon}\right)] = 0$$

Thus $\mathbb{E}_{\epsilon}[R_{i,j}] = \delta$ $\qquad\square$

## F. Proof of Theorem 4.2

*Proof.* Since $P_R(t) = \frac{P_W(t)}{\sum_{t\in\hat{R}}P_W(t)+\sum_{t\in\hat{G}}e^{-\hat{\delta}}P_W(t)}, t\in\hat{R}$, combining with the definition of $P_W$ (Equation 1), we have

$$P_R(t) = \frac{P(t)}{\sum_{t\in\hat{R},t\in R}P(t) + \sum_{t\in\hat{R},t\in G}e^{\delta}P(t) + \sum_{t\in\hat{G},t\in R}e^{-\hat{\delta}}P(t) + \sum_{t\in\hat{G},t\in G}e^{\delta-\hat{\delta}}P(t)}, t\in\hat{R},$$

analogously, we have

$$P_R(t) = \frac{e^{-\hat{\delta}}P(t)}{\sum_{t\in\hat{R},t\in R}P(t) + \sum_{t\in\hat{R},t\in G}e^{\delta}P(t) + \sum_{t\in\hat{G},t\in R}e^{-\hat{\delta}}P(t) + \sum_{t\in\hat{G},t\in G}e^{\delta-\hat{\delta}}P(t)}, t\in\hat{G},$$

Denoted by $A := \sum_{t\in\hat{R},t\in R}P(t) + \sum_{t\in\hat{R},t\in G}e^{\delta}P(t) + \sum_{t\in\hat{G},t\in R}e^{-\hat{\delta}}P(t) + \sum_{t\in\hat{G},t\in G}e^{\delta-\hat{\delta}}P(t)$, we have

$$|P_R(t) - P(t)| = \begin{cases} |\frac{1}{A} - 1|P(t), & t\in\hat{R} \text{ and } t\in R; \\ |\frac{e^{\delta}}{A} - 1|P(t), & t\in\hat{R} \text{ and } t\in G, \\ |\frac{e^{-\hat{\delta}}}{A} - 1|P(t), & t\in\hat{G} \text{ and } t\in R, \\ |\frac{e^{\delta-\hat{\delta}}}{A} - 1|P(t), & t\in\hat{G} \text{ and } t\in G, \end{cases} \tag{7}$$

Since

$$A = \sum_{t\in\hat{R},t\in R}P(t) + \sum_{t\in\hat{R},t\in G}e^{\delta}P(t) + \sum_{t\in\hat{G},t\in R}e^{-\hat{\delta}}P(t) + \sum_{t\in\hat{G},t\in G}e^{\delta-\hat{\delta}}P(t)$$
$$\geq \min\{1, e^{\delta-\hat{\delta}}\}\left(\sum_{t\in\hat{R},t\in R}P(t) + \sum_{t\in\hat{G},t\in G}P(t)\right) + e^{-\hat{\delta}}\left(\sum_{t\in\hat{R},t\in G}P(t) + \sum_{t\in\hat{G},t\in R}P(t)\right) \tag{8}$$
$$\geq \min\{1, e^{\delta-\hat{\delta}}\}(1-\epsilon_1) + e^{-\hat{\delta}}\epsilon_1 \geq e^{-\epsilon_2}(1-\epsilon_1) + e^{-\hat{\delta}}\epsilon_1,$$

and

$$A = \sum_{t\in\hat{R},t\in R}P(t) + \sum_{t\in\hat{R},t\in G}e^{\delta}P(t) + \sum_{t\in\hat{G},t\in R}e^{-\hat{\delta}}P(t) + \sum_{t\in\hat{G},t\in G}e^{\delta-\hat{\delta}}P(t)$$
$$\leq \max\{1, e^{\delta-\hat{\delta}}\}\left(\sum_{t\in\hat{R},t\in R}P(t) + \sum_{t\in\hat{G},t\in G}P(t)\right) + e^{\delta}\left(\sum_{t\in\hat{R},t\in G}P(t) + \sum_{t\in\hat{G},t\in R}P(t)\right) \tag{9}$$
$$\leq \max\{1, e^{\delta-\hat{\delta}}\}(1-\epsilon_1) + e^{\delta}\epsilon_1 \leq e^{\epsilon_2}(1-\epsilon_1) + e^{\delta}\epsilon_1,$$

we have

$$\max\{|\frac{1}{A} - 1|, |\frac{e^{\delta-\hat{\delta}}}{A} - 1|\} \leq \max\{\frac{e^{2\epsilon_2}}{1-\epsilon_1} - 1, 1 - \frac{e^{-2\epsilon_2}}{(1-\epsilon_1) + e^{\delta-\epsilon_2}\epsilon_1}\}$$

and

$$\max\{|\frac{e^{\delta}}{A} - 1|, |\frac{e^{-\hat{\delta}}}{A} - 1|\} \leq \max\{\frac{e^{\delta+2\epsilon_2}}{1-\epsilon_1} - 1, 1 - \frac{e^{-\delta-2\epsilon_2}}{(1-\epsilon_1) + e^{\delta-\epsilon_2}\epsilon_1}\}.$$

It is easy to observe that $\max\{|\frac{1}{A}-1|, |\frac{e^{\delta-\hat{\delta}}}{A}-1|\} \leq \max\{|\frac{e^{\delta}}{A}-1|, |\frac{e^{-\hat{\delta}}}{A}-1|\}$ since $\delta, \hat{\delta} > 0$.

Thus, denoted by $f_1(\epsilon_1, \epsilon_2) := \max\{\frac{e^{2\epsilon_2}}{1-\epsilon_1}-1, 1-\frac{e^{-2\epsilon_2}}{(1-\epsilon_1)+e^{\delta-\epsilon_2}\epsilon_1}\}$ and $f_2(\epsilon_1, \epsilon_2) := \max\{\frac{e^{\delta+2\epsilon_2}}{1-\epsilon_1}-1, 1-\frac{e^{-\delta-2\epsilon_2}}{(1-\epsilon_1)+e^{\delta-\epsilon_2}\epsilon_1}\}$,

$$
\begin{aligned}
\mathbb{TV}(P_R, P) &= \sum_t |P_R(t) - P(t)| \\
&= |\frac{1}{A}-1| \sum_{t\in\hat{R}, t\in R} P(t) + |\frac{e^{\delta-\hat{\delta}}}{A}-1| \sum_{t\in\hat{G}, t\in G} P(t) \\
&\quad + |\frac{e^{\delta}}{A}-1| \sum_{t\in\hat{R}, t\in G} P(t) + |\frac{e^{-\hat{\delta}}}{A}-1| \sum_{t\in\hat{G}, t\in R} P(t) \\
&\leq \max\{|\frac{1}{A}-1|, |\frac{e^{\delta-\hat{\delta}}}{A}-1|\}( \sum_{t\in\hat{R}, t\in R} P(t) + \sum_{t\in\hat{G}, t\in G} P(t)) \\
&\quad + \max\{|\frac{e^{\delta}}{A}-1|, |\frac{e^{-\hat{\delta}}}{A}-1|\}( \sum_{t\in\hat{R}, t\in G} P(t) + \sum_{t\in\hat{G}, t\in R} P(t)) \\
&\leq \max\{|\frac{1}{A}-1|, |\frac{e^{\delta-\hat{\delta}}}{A}-1|\}(1-\epsilon_1) + \max\{|\frac{e^{\delta}}{A}-1|, |\frac{e^{-\hat{\delta}}}{A}-1|\}\epsilon_1 \\
&\leq (1-\epsilon_1)\max\{\frac{e^{2\epsilon_2}}{1-\epsilon_1}-1, 1-\frac{e^{-2\epsilon_2}}{(1-\epsilon_1)+e^{\delta-\epsilon_2}\epsilon_1}\} \\
&\quad + \epsilon_1 \max\{\frac{e^{\delta+2\epsilon_2}}{1-\epsilon_1}-1, 1-\frac{e^{-\delta-2\epsilon_2}}{(1-\epsilon_1)+e^{\delta-\epsilon_2}\epsilon_1}\} \\
&= (1-\epsilon_1)f_1(\epsilon_1, \epsilon_2) + \epsilon_1 f_2(\epsilon_1, \epsilon_2).
\end{aligned}
\tag{10}
$$

$\square$

## G. Broader Impact

Machine learning models are transforming various sectors by improving efficiencies and tackling complex problems. Despite these beneficial impacts, there are valid concerns regarding the integrity and security of machine learning implementations (Wang et al., 2023d;c; Wu et al., 2022; 2023a;c; Mai et al., 2023b;a; Guo et al., 2024). In this context, watermarking has become a critical tool. It not only verifies the authenticity and ownership of digital media but also aids in identifying AI-generated content.

This paper contributes to the ongoing discussion by delving into advanced watermark removal techniques and introducing DE-MARK, a comprehensive framework designed for both stealing and removing n-gram-based watermarks. Our method addresses shortcomings of prior research by eliminating the need for specific knowledge about the underlying hash functions and avoiding the use of paraphrasing tools that disrupt the original LM distribution. Instead, DE-MARKemploys a querying strategy that allows for the accurate reconstruction of watermark parameters and ensures the integrity of the LM's output remains intact post-removal, offering a theoretical guarantee on the preservation of the original distribution.

While DE-MARKprovides a robust tool against watermark tampering, its capabilities also introduce potential risks. The ability to reverse-engineer watermarking processes could be misused to strip protective marks from proprietary or sensitive content, raising ethical concerns about the security and control of digital information. Additionally, the adaptability of DE-MARKfor watermark exploitation—where it can be used to generate watermarked content by mimicking original rules—further complicates the landscape of digital rights management.

