# OpenReview forum: "De-mark: Watermark Removal in Large Language Models"
_ICML.cc/2025/Conference — ICML 2025 poster_

### Official Review · Reviewer_kzSZ · 2025-03-09

**Overall Recommendation:** 2

**Summary:**

DE-MARK presents a framework for removing n-gram-based watermarks, specifically targeting the soft watermarking scheme proposed by Kirchenbauer et al. (2023a). The method utilizes a novel querying strategy called "random selection probing" to estimate watermark parameters like strength and red-green lists. The paper claims theoretical guarantees for distribution preservation and demonstrates the efficacy of DE-MARK on models like Llama3 and ChatGPT in watermark removal and exploitation tasks. The core contribution is a practical approach to reverse-engineer and remove a specific type of statistical watermark without prior knowledge of watermark parameters.

**Claims And Evidence:**

The contribution of this work appears limited.

Firstly, the attack is primarily focused on a specific soft watermark (Kirchenbauer et al., 2023a). This type of watermark, as demonstrated by Gu et al. (2024) in "On the Learnability of Watermarks for Language Models" (ICLR 2024), is known to be relatively easily learned. While the paper introduces refined techniques to estimate the hyperparameters of this soft watermark, the overall contribution in addressing a fundamentally weak watermark is arguably incremental. Secondly, the paper's scope does not extend to other watermarking methods, such as advanced cryptographic watermarks, which are designed to be theoretically resistant to parameter learning. Examples like "Undetectable watermarks for language models" highlight this gap.

Furthermore, the second contribution regarding industry-scale applicability is questionable. It is unknown whether ChatGPT employs watermarking, and if so, which specific technique. Simulating a soft watermark on top-20 tokens and demonstrating its removal with DE-MARK does not convincingly demonstrate effectiveness against a real-world, industry-level watermarking implementation.

Therefore, it is recommended that the authors revise the title to accurately reflect the limited scope of their method, specifically its applicability to soft n-gram watermarks. The current title is overly broad and implies a capability to remove a wider range of watermarks than is actually demonstrated.

**Essential References Not Discussed:**

Gu et al. (2024) "On the Learnability of Watermarks for Language Models" (ICLR 2024)

**Experimental Designs Or Analyses:**

Yes

**Methods And Evaluation Criteria:**

As state in "Claims And Evidence"

**Other Comments Or Suggestions:**

None

**Other Strengths And Weaknesses:**

None

**Questions For Authors:**

None

**Relation To Broader Scientific Literature:**

Related

**Theoretical Claims:**

Yes

---

> ### Author Rebuttal · Authors · 2025-04-01
>
> We thank the reviewer for the constructive feedback, which has been invaluable in refining our manuscript. Below, we provide detailed responses to each of your comments:
>
> > Q1: Firstly, the attack is primarily focused on a specific soft watermark (Kirchenbauer et al., 2023a). This type of watermark, as demonstrated by Gu et al. (2024) in "On the Learnability of Watermarks for Language Models" (ICLR 2024), is known to be relatively easily learned. While the paper introduces refined techniques to estimate the hyperparameters of this soft watermark, the overall contribution in addressing a fundamentally weak watermark is arguably incremental. Secondly, the paper's scope does not extend to other watermarking methods, such as advanced cryptographic watermarks, which are designed to be theoretically resistant to parameter learning. Examples like "Undetectable watermarks for language models" highlight this gap.
>
> A1: Our method is specifically designed for n-gram-based approaches, which are widely employed in watermarking[1,2,3,4]. Consequently, it can be naturally extended to other n-gram watermarking methods.
>
> To support our claim, we extend our method on two additional n-gram-based advanced distortion-free watermarking: $\gamma$-reweight[3] and DiPmark[4], the results are presented in Table 13 and 14 in this [anonymous link](https://docs.google.com/document/d/e/2PACX-1vSfxtMpq2yL7QjOjW0NNWgI_J4LG9QHes7eBtj4P7LqdrIVBTuibloz0p0LLG5dhijwS7UhFcVfw537/pub), which demonstrate Demark’s strong generalization capabilities beyond just the KGW framework.
>
> We are not able to evaluate our attack method on Undetectable watermarks because: a) it is not n-gram-based, b) the vocabulary size of the LM in their paper is only 2, and it is not trivial to extend their method to models with larger vocabulary size c) it’s a pure theoretical paper without any code implementation.
>
>
>
> > Q2: Furthermore, the second contribution regarding industry-scale applicability is questionable. It is unknown whether ChatGPT employs watermarking, and if so, which specific technique. Simulating a soft watermark on top-20 tokens and demonstrating its removal with DE-MARK does not convincingly demonstrate effectiveness against a real-world, industry-level watermarking implementation.
>
>
> A2: Whether ChatGPT employs a watermark by default is irrelevant to our study. In our experiments, we manually embedded a soft watermark into ChatGPT outputs and successfully removed it using Demark. Moreover, as noted in various prior works [1,2,3,4], the choice of LLM has minimal impact on the effectiveness of watermarking techniques. Our extensive results across LLaMA 3B, LLaMA 8B, and Mistral 7B further confirm that Demark is robust to the choice of underlying LLM. Finally, both Reviewer 5v1A and eS2n acknowledged that our case study on ChatGPT is informative and valuable.
>
>
> > Q3: Therefore, it is recommended that the authors revise the title to accurately reflect the limited scope of their method, specifically its applicability to soft n-gram watermarks. The current title is overly broad and implies a capability to remove a wider range of watermarks than is actually demonstrated.
>
>
> A3: Our method is designed for n-gram-based approaches and is applicable beyond just soft n-gram watermarks, as demonstrated by our experiments in Tables 13 and 14, available in this [anonymous link](https://docs.google.com/document/d/e/2PACX-1vSfxtMpq2yL7QjOjW0NNWgI_J4LG9QHes7eBtj4P7LqdrIVBTuibloz0p0LLG5dhijwS7UhFcVfw537/pub). In response to the reviewer's recommendation, we will revise the title to emphasize our focus on n-gram watermarking.
>
> We would be more than happy to further discuss any additional questions or concerns you may have.
>
>
> [1] A watermark for large language models, Kirchenbauer et al., ICML 2023
> [2] On the reliability of watermarks for large language models, Kirchenbauer et al., ICRL 2024
> [3] Unbiased Watermark for Large Language Models, Hu et al., ICLR 2024
> [4] DiPmark: A Stealthy, Efficient and Resilient Watermark for Large Language Models, Wu et al., ICML 2024

---

### Official Review · Reviewer_eS2n · 2025-03-14

**Overall Recommendation:** 3

**Summary:**

## Summary

This paper addresses critical vulnerabilities in n-gram watermarking schemes for language models (LMs). The authors propose **DE-MARK**, a framework for watermark removal and exploitation, with three key contributions:

1. **Watermark Parameter Estimation**
   Introduces _random selection probing_ to reconstruct red-green token lists and estimate watermark strength \\(\delta\\) without prior knowledge of hash functions or n-gram parameters. Core algorithms include:
   - Relative probability ratio matrix
   - Context length detection via score consistency
   - Provable unbiased \\(\delta\\) estimation

2. **Distribution-Preserving Removal**
   Formulates watermark removal as probability reweighting
   Theoretical guarantees bound the total variation distance between post-removal (\\(P_R\\)) and original (\\(P\\)) distributions

3. **Empirical Validation**
   Tests on Llama3, Mistral, and ChatGPT show:
   - **Removal**: Reduces TPR@FPR=0.1% from >89% to <15% while preserving GPT quality scores (\\(\Delta<0.5\%\\))
   - **Exploitation**: Achieves 93% TPR@FPR=0.1% when implanting stolen watermarks
   - **Black-box Adaptation**: 85% precision in red-green list detection with 10 samples

The work demonstrates fundamental limitations in current n-gram watermark robustness and provides tools for security auditing of LM watermarking systems.

## Update after Rebuttal
I found that the authors have adequately solved most of my concerns, and I recognize the contribution of the paper and would keep my rating as "weak accept".

**Claims And Evidence:**

**Well-supported claims:**

The paper provides detailed descriptions of the DE-MARK framework through five well-formulated algorithms covering relative probability calculation, token scoring, n-gram length identification, watermark strength estimation, and green list identification.

DE-MARK significantly reduces watermark detectability (TPR@FPR) from high levels (>60%) to low levels (<20%) while maintaining text quality as measured by GPT scores.

**Weaknesses in evidence:**

The experimental results don't include variance measures or statistical significance tests across multiple runs.

**Essential References Not Discussed:**

N/A

**Experimental Designs Or Analyses:**

The experiment settings and dataset usage are sound but need computational efficiency analysis and statistical significance testing.

**Methods And Evaluation Criteria:**

The paper's approach is methodologically sound:
- Random selection probing efficiently estimates token probabilities
- Five sequential algorithms form a coherent framework for watermark removal
- Theoretical guarantees establish bounds on distribution gaps

## Evaluation Criteria Assessment

- Diverse models (Llama3.1-8B, Mistral-7B, Llama3.2-3B) and datasets (Dolly, MMW)
- Appropriate metrics (TPR@FPR, p-values, GPT scores)

**Weaknesses:**
- Missing computational efficiency analysis
- Lack of statistical significance testing

**Other Comments Or Suggestions:**

I have known that the setting of this method is stronger than priors, but I also want to see the performance comparisons with others, e.g., `Attacking llm watermarks by exploiting their strengths`, which could make this paper more sound.

**Other Strengths And Weaknesses:**

Strengths:
The setting of this paper is strong enough, detector-free scenarios are genuinely in the real world.

The theorems and proofs are sound, making the paper good enough.

Weaknesses:

I would like to see the analysis of the number of queries.

Some formulas need to be polished in formats.

Some references should be referred to correctly, e.g., `Watermark stealing in large language models`  should be in ICML.

In section 4, the authors should point out that detailed proofs of theorems are in Appendix XXX.

**Questions For Authors:**

In section 5.2, why only consider $h=\{3,7\}$? In normal practices, $h=2$ is normally used, and I would like to see a detailed comparison between the selection of $h$.

Theorem 4.2 provides bounds on distribution gaps, but there's no experimental validation of how tight these bounds are in practice. Can you provide empirical measurements of actual distribution gaps compared to the theoretical bounds?

**Relation To Broader Scientific Literature:**

The contribution of this paper is unique compared to broader scientific literature, which are introduced in the related work part.

**Theoretical Claims:**

I have checked the theoretical claims of the paper, especially theorems in section 4, which are sound.

---

> ### Author Rebuttal · Authors · 2025-04-01
>
> Thank you for your valuable feedback and thoughtful suggestions, which have greatly helped improve the quality of our work.
>
> > Q1 The experimental results don't include variance measures or statistical significance tests across multiple runs.
>
> A1: Firstly, evaluating whether a sentence is watermarked does not require running multiple statistical significance tests because the watermark detection algorithm itself is deterministic and already employs a statistical hypothesis test [1]. Specifically, during detection, we set a theoretical false positive rate (e.g., 0.1%), calculate the corresponding statistical threshold, and then flag the sentence as watermarked if its score exceeds this threshold. To the best of our knowledge, most watermarking papers[1,2,3,4] do not perform multiple-run tests, and we have adopted similar settings.
>
> Moreover, we provide results across multiple datasets and LLMs to mitigate the effects of randomness.
>
>
> > Q2 Missing computational efficiency analysis/analysis of the number of queries.
>
> A2: Please refer to Response A2 in our rebuttal to reviewer 5v1A for a detailed discussion on query efficiency.
>
> > Q3 Some formulas need to be polished in formats. Some references should be referred to correctly
>
> Thank you for your valuable suggestion. We sincerely apologize for the mistake. We will revise the formatting and correct the references accordingly in the version.
>
> > Q4 In section 4, the authors should point out that detailed proofs of theorems are in Appendix XXX.
>
> We apologize for the confusion. In the revision, we will clearly link the proofs of the theorems within the main body of the text.
>
> > Q5 Comparision against Attacking llm watermarks by exploiting their strengths
>
> The setting of our work fundamentally differs from that of Pang, Qi, et al. [5], which represents an early exploration of watermark removal under relatively simple assumptions. Specifically, their attacks are limited to the following scenarios:
>
> - Naive token editing (Sec. 4): This method inevitably leads to degraded text quality.
>
> - Multi-key scenario (Sec. 5): This is a trivial case where a simple averaging technique suffices, in contrast to our focus on the more practical and challenging one-key scenario.
>
> - Detector-available scenario (Sec. 6): Prior studies, including this one, have shown that attacks in this setting are relatively easy. In contrast, our work addresses the more difficult detector-unavailable scenario (D0 setting, detailed in Sec. 3.1, Lines [138–143]).
>
> Therefore, we believe a direct comparison would be neither meaningful nor informative.
>
> > Q6 I would like to see a detailed comparison between the selection of h
>
> We kindly note that additional results regarding the selection of $h$ are provided in the Appendix Figure 6 and Figure 7.
>
> In response to the recommendation, we have also included results for identifying the prefix n-gram length with $h$ = 2, 4, 6 and 8 in Figures 9–10, available via this [anonymous link](https://docs.google.com/document/d/e/2PACX-1vSfxtMpq2yL7QjOjW0NNWgI_J4LG9QHes7eBtj4P7LqdrIVBTuibloz0p0LLG5dhijwS7UhFcVfw537/pub).
>
> The results show that our method generalizes well across different selections of $h$.
>
> > Q7 Theorem 4.2 provides bounds on distribution gaps, but there's no experimental validation of how tight these bounds are in practice. Can you provide empirical measurements of actual distribution gaps compared to the theoretical bounds?
>
> We empirically evaluated our method on 3,000 token distributions from the MMW Book Report dataset using the LLaMA3.2 3B model. Our results show that $\mathbb{E}[\frac{\mathbb{TV}(P_R,P)}{\epsilon_1 f_2(\epsilon_1,\epsilon_2)+ (1-\epsilon_1) f_1(\epsilon_1,\epsilon_2)}]=0.328$ indicating that the distribution gap is well-controlled by the bound.
>
> Moreover, our GPT score evaluation results in Table 1 and Table 2 further confirm that the generated text remains highly consistent with the original in distribution, demonstrating minimal distribution distortion and strong preservation of text quality.
>
>
> [1] A watermark for large language models, Kirchenbauer et al., ICML 2023
> [2] Robust Distortion-free Watermarks for Language Models, Kuditipudi et al., TMLR 2023
> [3] Unbiased Watermark for Large Language Models, Hu et al., ICLR 2024
> [4] DiPmark: A Stealthy, Efficient and Resilient Watermark for Large Language Models, Wu et al., ICML 2024
> [5] Attacking LLM Watermarks by Exploiting Their Strengths, Pang et al, ICLR 2024 Workshop

---

> > ### Comment · Reviewer_eS2n · 2025-04-05
> >
> > Thank you to the authors for the thorough and thoughtful response. Your clarifications have addressed nearly all of my concerns, particularly around statistical testing, query efficiency, and theoretical assumptions.
> >
> > Regarding concerns from other reviewers about scope: while the proposed method focuses on n-gram-based watermarking, this is still a meaningful contribution. The KGW watermark remains one of the most influential schemes in the field, and providing a principled and effective attack against it—both theoretically and empirically—is valuable. I hope other reviewers will take this into account when evaluating the contribution.

---

> > > ### Author Response · Authors · 2025-04-06
> > >
> > > Thank you very much for your thoughtful and encouraging feedback. We truly appreciate your recognition of the novelty and value of our work, and we are glad that our clarifications effectively addressed your concerns. Thank you again for your time and constructive engagement.

---

### Official Review · Reviewer_5v1A · 2025-03-15

**Overall Recommendation:** 3

**Summary:**

- The paper presents DE-MARK, a framework designed to remove n-gram-based watermarks from Large Language Models (LLMs). It introduces a novel querying strategy, "random selection probing," to assess watermark strength and reconstruct watermarking parameters.
- Unlike previous methods that rely on knowledge of the watermarking function or require paraphrasing, DE-MARK offers a general approach that estimates watermark parameters and reverses the watermarking effects while preserving the original language model’s distribution.
- Beyond removal, DE-MARK can also be used to exploit watermarks by mimicking their structure, effectively generating watermarked text using an attacker LLM, which raises concerns about the security of watermarking scheme
- The paper demonstrates the effectiveness of DE-MARK in both watermark removal and exploitation tasks on models such as Llama3 and ChatGPT, achieving a significant drop in watermark detectability without degrading text quality.


### Nits and Prior Relevant Work Which has not been cited:

- Line 102: However, The -> However, the

**Claims And Evidence:**

- The paper demonstrates that DE-MARK significantly reduces watermark detectability in models like Llama3 and Mistral. After applying DE-MARK, the true positive rate (TPR) of watermark detection drops from over 60-90% to below 20%, even at low false positive rates. The authors also provide theoretical guarantees that the post-removal distribution remains close to the original model distribution. (Table 1)
- Unlike previous methods that require access to the underlying hash function or paraphrasing tools, DE-MARK estimates watermark parameters (like red-green token lists and watermark strength) through a novel querying strategy called random selection probing. This approach allows it to infer the watermark without direct access to its configuration.
- The paper shows that after learning the watermarking pattern, an adversary can generate text that mimics a watermarked LLM. Experiments on watermark exploitation confirm that DE-MARK can recreate watermarked content with high accuracy, making it possible to generate fake watermarked text.
- The paper uses the GPT score as a metric to assess text quality before and after watermark removal. Results show that even after applying DE-MARK, the GPT score remains nearly the same, indicating that the method does not degrade fluency, coherence, or correctness in the generated text.

**Essential References Not Discussed:**

None

**Experimental Designs Or Analyses:**

- The paper evaluates DE-MARK on multiple language models, including Llama3 (3B & 8B), Mistral 7B, and ChatGPT.
- This variety ensures that the method is not tied to a specific model architecture and can generalize across different LLMs.
- The authors measure watermark detectability before and after applying DE-MARK using true positive rate (TPR) at fixed false positive rates (FPR).
- Results show that after DE-MARK is applied, watermark detectability drops significantly, indicating successful removal.
- The paper also evaluates the impact of DE-MARK on text quality using GPT-score, showing that the removal process does not significantly degrade generation quality.

**Methods And Evaluation Criteria:**

- The paper does a good job of explaining why watermark removal is an important and underexplored area. The motivation is clear, and the threat model is well-defined.
- Unlike previous work that relies on paraphrasing or explicit knowledge of the watermarking function, DE-MARK estimates the watermark structure through a statistical approach. The authors also provide a theoretical bound on the difference between the original and post-removal distributions, which strengthens the credibility of their method.
- The authors evaluate DE-MARK on multiple models, including Llama3 and Mistral, and even apply it to ChatGPT in a real-world case study. The variety of models tested helps support the claim that DE-MARK generalizes well across different LLMs
- The evaluation considers both watermark detectability (TPR@FPR) and text quality (GPT score). This shows not only how well DE-MARK removes watermarks but also that it does not degrade output quality.

**Other Comments Or Suggestions:**

- Addressing practical concerns I raised above such as scalability, robustness to different watermarking techniques, and real-world constraints (e.g., no probability access) would make the paper even stronger. If these aspects were explored further, the work would have a broader impact and provide even more value to the AI research community.

**Other Strengths And Weaknesses:**

- The paper focuses on n-gram-based watermarks, but modern watermarking techniques are evolving. Would DE-MARK still work against semantic watermarks or distortion-free watermarking? Even if not tested, discussing this in the limitations section would make the paper stronger.
- The method requires multiple queries to reconstruct the watermarking scheme, which could be computationally expensive. It would be helpful if the paper included an analysis of query efficiency—how many queries are typically needed to achieve successful watermark removal? Could optimizations reduce this cost?
- While DE-MARK is tested on ChatGPT, the paper does not extensively evaluate settings where log probabilities are entirely unavailable (a common scenario for many commercial LLM APIs).
- The assumption about gaussian noise also seems like a strong and arbitrary assumption to me without any empirical evidence.
- In addition to the unbiasedness of the estimate it would also be helpful to do a consistency analysis of the estimator i.e. how fast does the estimate converge? How many queries are required to decode a watermark and the practical feasibility of DEMARK.

**Questions For Authors:**

One thing that isn’t super clear to me is that it seems this watermark removal assumes a KGW watermarking scheme. There are many more watermarking schemes which are also n-gram watermarking schemes but don't explicitly add a delta to the logit params and also have distortion free property such as the Gumbel watermark by Aaronson. Would this watermark removal scheme also work on that? If not, I suggest authors to tone down the claims in the paper. Open to more discussion on this.

**Relation To Broader Scientific Literature:**

1. The paper builds on prior work in statistical watermarking for LLMs (e.g., Kirchenbauer et al. (2023)), which proposed methods to embed watermarks into AI-generated text for provenance tracking.
2. It directly addresses recent security concerns raised by Jovanović et al. (2024) on the feasibility of watermark stealing and removal attacks.
3. By providing a provable theoretical bound on watermark removal effectiveness, DE-MARK contributes a formal framework to this discussion, which is a notable improvement over purely empirical attack demonstrations.

**Theoretical Claims:**

- The authors derive an upper bound on the total variation distance between the original and post-removal language model distributions. This is an important result because it provides a theoretical guarantee that DE-MARK does not significantly alter the statistical properties of the model’s output.
- The bound explicitly depends on two key error terms: (1) misclassification error $(\epsilon_1)$, which accounts for incorrectly identified watermark tokens, and (2) $estimation error (\epsilon_2)$, which quantifies how accurately DE-MARK estimates the watermark strength.
- The proof in the appendix looks correct based on working out the steps myself though more details could be added to make it easier to follow.

---

> ### Author Rebuttal · Authors · 2025-04-01
>
> > Q1: Would DE-MARK still work against modern watermarking techniques (e.g. distortion-free methods like Gumbel watermark by Aaronson)? Is it specially designed for KGW?
>
> A1: Our proposed methods can be generalized to most n-gram-based methods, and we add additional experiments for two popular distortion-free n-gram-based watermarking algorithms to support our claim, $\gamma$-reweight[1] and DiPmark[2]. Please see Table 13 and Table 14 in this [anonymous link](https://docs.google.com/document/d/e/2PACX-1vSfxtMpq2yL7QjOjW0NNWgI_J4LG9QHes7eBtj4P7LqdrIVBTuibloz0p0LLG5dhijwS7UhFcVfw537/pub). The results show that our method can effectively remove the watermarks.
>
> However, it is not possible to remove the Gumbel watermark, as it only produces a one-hot probability distribution, i.e., assigning a probability of 1 to a single token and 0 to all others. This output lacks sufficient information to theoretically bound the difference between the original and post-removal distributions of the language model.
>
> > Q2: Analysis of query efficiency?
>
> A2: We first present a detailed time efficiency analysis to show that our time cost is acceptable, then we show some acceleration methods to further reduce the time cost.
>
> **Time Efficiency Analysis**
>
> The experiments were conducted on a single RTX 6000 Ada GPU. Table 11 and Table 12 in this [anonymous link](https://docs.google.com/document/d/e/2PACX-1vSfxtMpq2yL7QjOjW0NNWgI_J4LG9QHes7eBtj4P7LqdrIVBTuibloz0p0LLG5dhijwS7UhFcVfw537/pub) present the time efficiency of watermark stealing and removal with respect to varying numbers of queries.
>
> For De-mark, we require $m(m - 1)$ queries per token. In our experiments, we set $m = 20$, resulting in a total of 380 queries per token. While this number may seem high, it is important to note that repeated querying is a necessary strategy to comprehensively gather the required information.
>
> Despite this, our experimental results indicate that the time cost is acceptable when compared to the baseline. The computation remains efficient due to the following reasons:
>
> - Each query is very short;
> - The query templates are identical across inputs, allowing for the reuse of most pre-computed key-value pairs, which significantly accelerates computation;
>
> **Acceleration method**
>
> For watermark identification, the time cost is already low, and we find no additional optimization necessary.
>
> For watermark removal and exploitation, we can achieve 2$\times$ to 4$\times$ speedup by randomly removing token pairs in Alg. 1. This introduces a trade-off between removal performance and inference speed. Additionally, increasing the value of $\eta$ can yield improved outcomes, the results are presented in Table 15 in the [anonymous link](https://docs.google.com/document/d/e/2PACX-1vSfxtMpq2yL7QjOjW0NNWgI_J4LG9QHes7eBtj4P7LqdrIVBTuibloz0p0LLG5dhijwS7UhFcVfw537/pub).
>
> A more sophisticated token pair clipping strategy may also lead to further performance improvements.
>
> > Q3: The paper does not extensively evaluate settings where log probabilities are entirely unavailable (a common scenario for many commercial LLM APIs).
>
> A3: Regarding watermark identification and exploitation, we kindly point out that we have conducted extensive experiments in the black-box setting (i.e., where log probabilities are entirely unavailable), as shown in Tables 2, 3, and 9, and Figures 4, 5, and 7.
>
> As for watermark removal, as discussed in Section 3.1, when log probabilities are completely unavailable, it is impossible to bound the gap between the original and post-removal distributions of the language model. This is because the output provides only a single token, which is insufficient to recover the full distribution.
>
> > Q4: The assumption about Gaussian noise also seems like a strong and arbitrary assumption
>
> A4: Thank you for pointing this out. Actually, in Appendix E. Proof of Theorem 4.1, the gaussian noise assumption is not necessary for the proof, we only require the noise to be symmetrically distributed (i.e., $P(\epsilon=x)=P(\epsilon=-x),\forall x \in \mathbb{R}$), which is a mild and intuitive assumption. We will update the assumption accordingly in our revision.
>
> > Q5: How fast does the estimate converge? How many queries are required to decode a watermark
>
> A5: Demark has relatively low computational cost, which enables us to even increase the number of queries for more accurate estimations. Please refer to Figure 8 in this [anonymous link](https://docs.google.com/document/d/e/2PACX-1vSfxtMpq2yL7QjOjW0NNWgI_J4LG9QHes7eBtj4P7LqdrIVBTuibloz0p0LLG5dhijwS7UhFcVfw537/pub) for the convergence speed of estimating $\delta$. We evaluate the performance using different values of $m$ in Algorithm 4 and vary the parameter $c$ (set to 1, 2, 5, 10, 20, 50, and 100) to generate different numbers of queries.
>
> [1] Unbiased Watermark for Large Language Models, Hu et al., ICLR 2024
> [2] DiPmark: A Stealthy, Efficient and Resilient Watermark for Large Language Models, Wu et al., ICML 2024

---

### Decision · Program_Chairs · 2025-05-01

**Decision:**

Accept (poster)

**Comment:**

The paper proposes DE-MARK, a method to remove n-gram-based LLM watermarks. The key idea is based on a novel querying technique, random selection probing, to estimate watermark configuration parameters, including watermark strength and green-red token lists.  The paper claims theoretical guarantees for distribution preservation and demonstrates the efficacy of DE-MARK on models like Llama3 and ChatGPT in watermark removal and exploitation tasks.

Strength of the paper:
- The paper develops random selection probing technique to estimate watermark configuration parameters without prior knowledge.
- The paper demonstrates that DE-MARK significantly reduces watermark detectability in models like Llama3 and Mistral. After applying DE-MARK, the true positive rate (TPR) of watermark detection drops from over 60-90% to below 20%.
- The paper provides a theoretical bound on the effectiveness of watermark removal.

Weakness of the paper:
- There is already a prior study showing n-gram watermarks such as KGW is weak against attacks.
- Some experimental settings are weak, i.e.g simulating a soft watermark on top-20 tokens and demonstrating its removal with DE-MARK.
- Analysis of complexity and the number of probes is missing.
- The use of GPT to evaluate the generation quality is questionable. It is better to use a model-based metric such as COMET, BERTScore, MetricX, SEScore.